# Carrier transport theory for twisted bilayer graphene in the metallic regime

Gargee Sharma [1,2,10], Indra Yudhistira [1,3,10], Nilotpal Chakraborty[1,4,5], Derek Y. H. Ho[1,4], M. M. Al Ezzi[1,3], Michael S. Fuhrer [6,7], Giovanni Vignale [1,4,8] & Shaffique Adam [1,3,4,9 ✉]

Understanding the normal-metal state transport in twisted bilayer graphene near magic angle is of fundamental importance as it provides insights into the mechanisms responsible for the observed strongly correlated insulating and superconducting phases. Here we provide a rigorous theory for phonon-dominated transport in twisted bilayer graphene describing its unusual signatures in the resistivity (including the variation with electron density, temperature, and twist angle) showing good quantitative agreement with recent experiments. We contrast this with the alternative Planckian dissipation mechanism that we show is incompatible with available experimental data. An accurate treatment of the electron-phonon scattering requires us to go well beyond the usual treatment, including both intraband and interband processes, considering the finite-temperature dynamical screening of the electron-phonon matrix element, and going beyond the linear Dirac dispersion. In addition to explaining the observations in currently available experimental data, we make concrete predictions that can be tested in ongoing experiments.

[1] Centre for Advanced 2D Materials, National University of Singapore, 6 Science Drive 2, 117546 Singapore, Singapore. [2] School of Basic Sciences, Indian Institute of Technology Mandi, Mandi 175005, India. [3] Department of Physics, National University of Singapore, 2 Science Drive 3, 117551 Singapore, Singapore. [4] Yale-NUS College, 16 College Avenue West, 138527 Singapore, Singapore. [5] Max-Planck-Institut für Physik komplexer Systeme, Nöthnitzer Straße 38, Dresden 01187, Germany. [6] ARC Centre of Excellence in Future Low Energy Electronic Technologies, Monash University, Monash, VIC 3800, Australia. [7] School of Physics and Astronomy, Monash University, Monash, VIC 3800, Australia. [8] Department of Physics and Astronomy, University of Missouri, Columbia, MO 65211, USA. [9] Department of Materials Science and Engineering, National University of Singapore, 9 Engineering Drive 1, 117575 Singapore, Singapore. [10]These authors contributed equally: Gargee Sharma, Indra Yudhistira. ✉email: shaffique.adam@yale-nus.edu.sg

The seminal observation of superconductivity and correlated insulating states in twisted bilayer graphene (tBG)[1–3] has generated tremendous excitement in the physics community[4]. At present, there is no consensus on the mechanism responsible for these observations. It was anticipated almost 15 years ago[5] that when two sheets of graphene are stacked on top of each other with a slight relative rotation, a large wavelength moiré superlattice potential emerges. By reducing the twist angle in these moiré systems, the Bloch period can be increased by two orders of magnitude thereby bridging the lengthscales between naturally occurring lattices in materials and optical traps of cold atoms[6].

The addition of a moiré potential significantly modifies the underlying electronic structure, including both a reduction in the Fermi velocity at low energy and a reduction of the bandwidth of the lowest energy band. Both these effects enhance the importance of electron–electron interactions[7]. These properties can be understood as follows: In the absence of any coupling between the layers, the original Dirac-like bands are just folded onto the smaller moiré Brillouin zone as determined by symmetry, but not modified; it is the interlayer coupling that causes level repulsion between the folded moiré bands. The moiré band closest to charge neutrality remains Dirac-like at low energy (the sublattice symmetry protecting the Dirac cones is not broken by the moiré potential) but with a reduced Fermi velocity, and the first moiré band as a whole gets squeezed by the level repulsion. This reduced bandwidth is quantified by the separation in the energy of the two van Hove singularities (VHS) that are found at the midpoint between the two original (relatively rotated in the Brillouin zone) Dirac cones, but pushed closer in energy by the level repulsion. Numerical ab initio studies soon confirmed the predictions of this long-wavelength continuum picture[8]; however, the experimental situation remained controversial for a while (see e.g., ref. [9]). Since then, the continuum model has been largely confirmed experimentally (see e.g., refs. [10,11]).

Taking the continuum model to its logical conclusion, Bistritzer and MacDonald predicted[12] that the Fermi velocity would vanish at a family of so-called "magic angles". Their original work assumed that the lattices remained rigid. More recent work including lattice relaxation effects[13,14] suggests that only the first and largest magic angle ($\theta_M \sim 1.06°$) is stable, and that the rigid lattice continuum approximation breaks down for smaller angles.

It should be emphasized that within the continuum model, strictly speaking, the bandwidth or $2\varepsilon_{VHS}$ remains finite at the magic angle. However, experimentally, at least in local spectroscopy measurements (e.g., refs. [15–18]), an alternate definition of magic angle is possible, i.e., when $\varepsilon_{VHS} = 0$. These would occur at angles below the original magic angle and in the regime where lattice relaxation effects are dominant (and it is not clear, in this case, what the electronic structure would look like). Given the observation of strongly correlated physics in other twisted 2D materials[19–22], it seems that the vanishing bandwidth is more germane than the vanishing Fermi velocity, although, at present, the relation between the two has not been established.

In this work, we establish yet another special angle, $\theta_{cr} \sim 1.15°$, the angle at which the Fermi velocity equals the phonon velocity. We show that at this angle, the phonon contribution to the resistivity strictly vanishes, and the experimentally measured resistivity would increase by several orders of magnitude for small deviations in angle on either side of $\theta_{cr}$. It has become normative in this quickly evolving field to attribute factor of ~5 changes in the resistivity[23] as evidence for superconductivity, and our work suggests more caution. By construction, each magic angle must be accompanied by two critical angles $\theta_{cr}$ (above and below $\theta_M$), and therefore its effects should be robust to lattice relaxation effects (as we demonstrate explicitly below). We demonstrate that the Fermi velocity of the linear bands and the VHS at the edges of the moiré Brillouin zone have distinct effects on the resistivity, and these could therefore be used in transport experiments to disentangle the importance of each in the correlated regime.

The present work is not about the observed superconductivity or correlated insulators. As we explain here, there is a geometric enhancement of the electron–phonon coupling in such moiré systems[24,25] that would favor a phonon mechanism for superconductivity; however, in a separate paper[26] we show that plasmons are also strongly enhanced and that superconductivity can arise from a purely electronic mechanism. Similarly, at present, it is unclear if the correlated insulator is a Mott insulator (see e.g., refs. [27–35]) or a Wigner crystal[36,37]. Furthermore, it is possible for long-range interactions to significantly distort the noninteracting bands away from charge neutrality due to the formation of inhomogeneous electrostatic potentials (see e.g., refs. [38,39]) although the experiments seem to suggest otherwise.

Rather, this work is about the carrier transport theory in the metallic regime (including at the van Hove singularity at higher carrier densities). We find that the role of phonons in tBG is perhaps as interesting as that of electrons: the same moiré potential that gives rise to the flat electronic bands, also results in enhancement of the electron–phonon coupling. Soon after the first experiments[1–3], we predicted[40] that charged impurities would always dominate the resistivity at the lowest carrier densities and temperatures, but that gauge phonons would dominate for most of the experimental window. This crossover is also present in monolayer graphene, but occurs at a temperature of ~ 500 K, while for tBG the crossover happens at ~5 K. As we show in Methods, available experimental data largely confirm our earlier predictions.

By now there have been two experimental transport studies focusing on the metallic regime. The first is from the MIT group[41] and the second is a UCSB-Columbia collaboration[42]. While the two experiments are largely consistent with each other, they arrive at very different conclusions on the dominant scattering mechanisms at play. Ref. [41] argues for a Planckian mechanism to explain their data, which implies a scattering rate $\hbar\tau^{-1} = Ck_BT$, where $C \lesssim 1$[43]. Here $C = 1$ is the Planckian bound set by holography and believed to be relevant for strange metals[44]. They argue that the linear-in-temperature behavior persisting well below the Bloch–Gruneisen temperature and the saturation of resistivity at higher temperature are both inconsistent with the

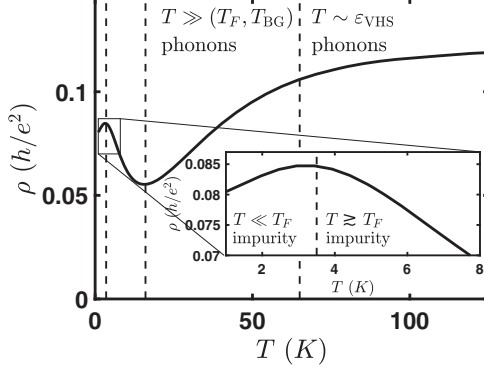

**Fig. 1 Illustration of the different carrier transport regimes in twisted bilayer graphene.** Electron–phonon scattering dominates except at very low temperature. At intermediate temperature ($(T_F, T_{BG}) \ll T < T_{VHS}$, resistivity is linear-in-$T$ and at high temperature ($T \sim \varepsilon_{VHS}$), resistivity saturates as a function of temperature. Inset: electron-impurity scattering dominates the transport at low temperature. Resistivity increases for $T \ll T_F$ and decreases for $T \gtrsim T_F$. This nonmonotonicity arises from the charged impurity scattering[40], and when the system crosses over from the degenerate to the nondegenerate regime.

conventional theory of phonon transport. We show here that both of these features are actually essential features of phonon-limited transport in tBG (see Fig. 1). While a microscopic theory showing Planckian dissipation has not yet been developed for tBG (it has for other systems, see e.g., ref. [45,46]), we can assume a Planckian mechanism to make predictions for the transport. We show here that a Planckian mechanism also gives a saturation in resistivity at high temperature and linear-in-temperature behavior at a low temperature consistent with experimental observations. In this work, we focus on the metallic regime that is far from the superconducting regime. A detailed analysis ultimately shows that the dominant scattering mechanism in this regime is not Planckian for several reasons including (a) the Planckian theory also predicts a strong carrier density dependence (absent in the experiment); (b) the experiment and the phonon mechanism both show the resistivity saturation at high temperature is set by the VHS energy, while for the Planckian theory this saturation is intrinsic (i.e., independent of bandstructure); (c) the twist angle dependence of Fermi velocity as extracted from experiment for phonon-limited scattering is consistent with the continuum theory[12–14], while it is orders of magnitude off for the Planckian theory; and most significantly, (d) the extracted value of the scattering time from the experiment using the Planckian theory contradicts the assumptions of the Planckian theory. Our work shows that the phonon interpretation of ref. [42] is consistent with the theory we develop here.

## Results and discussion

**Comparison of resistivities**. In Fig. 2 we compare data from ref. [42] (middle panels) with both a phonon-limited theory (left panel) and a Planckian theory (right panel). Similar to the experimental data, the phonon-mediated theory has weak density dependence. By contrast, the resistivity of the Planckian theory has strong density dependence (not seen in the experiment) that results from the density of states dependence of the Drude weight, which, unlike electron–phonon, remains uncompensated by the scattering time. We note that both the phonon-limited theory and the Planckian theory are linear-in-$T$ at low temperature, and saturate at high temperature (qualitatively similar to what is seen experimentally). However, the origin of the saturation is very different. For phonon scattering, the saturation is set by the electronic bandwidth $2\varepsilon_{VHS}$, while for Planckian dissipation it is mostly independent of $\varepsilon_{VHS}$ and set by Planckian strength $C$, which is expected to be somewhat universal and $C \leq 1$. This illustrates that both the phonon-limited theory and the Planckian theory provide robust predictions that can be tested against experiment. Near magic angle, tBG has electron-hole asymmetry that arises from the second-nearest hoping in the effective tight-binding model for graphene. The superlattice potential renormalizes the kinetic energy scales including the asymmetry to lower energies[12]. We include this effect in the theory by fitting separately for the electron and hole side (see below). Details of how we fit the experimental data to electron–phonon and Planckian theory are provided in Methods.

**Linear-in-$T$ resistivity**. Since phonons appear to dominate the transport properties, we are motivated to carefully consider the role of electron–phonon scattering. In this paper, we investigate the problem of normal state electronic transport in tBG focusing on the role of electron–phonon collision, which we find is the most important scattering mechanism in the relevant temperature and density regimes. The usual treatment of the electron–phonon does not take into account interband scattering, that we find below to be crucial near the magic angle. Secondly, dynamical screening of phonons is completely neglected because

typically $v_F \gg c_{ph}$, which again no longer holds true near the magic angle. Thirdly, geometric enhancement of the gauge phonon mode remains poorly addressed, which we find below to be the most dominant phonon mode in tBG. Lastly, the usual treatment of the electron–phonon problem has limited validity and fails beyond the linear regime (near VHS) that pushes us to go beyond the Dirac approximation and include nonlinear lattice effects. An accurate treatment of the electron–phonon scattering requires us to go well beyond the usual treatment, whereby including both interband and intraband processes, we show, for example, that the interband process allows for a linear-in-$T$ behavior well below the Bloch–Gruneisen temperature and the transition between the two is accompanied by several orders of magnitude decrease in the resistivity at a critical angle $\theta_{cr}$, distinct from the magic angle $\theta_M$. By considering the finite-temperature dynamical screening of the electron–phonon matrix element, we show, for example, that only the antisymmetric gauge phonon mode survives at a low twist angle; and by going beyond the linear Dirac dispersion, we show that the van Hove singularity causes saturation in resistivity as a function of temperature. In addition to explaining the observations in currently available experimental data, our theory makes concrete predictions that can be tested in ongoing experiments.

To our knowledge, the Boltzmann transport theory for acoustic phonon scattering in monolayer graphene was first developed by Hwang and Das Sarma[47]. In this work, we adopt the same formalism with several extensions appropriate for tBG. First, we consider both intraband and interband processes. As we show in the Supplementary Information, close to the magic angle, only interband scattering is operational which was not considered in ref. [47]. Second, while the linear Dirac Hamiltonian was an appropriate model for monolayer graphene, the reduced energies in tBG requires us to use an effective two-band Hamiltonian first proposed by ref. [48] that captures the physics near the van Hove singularity. Finally, we do the full finite-temperature and finite frequency RPA screening of the electron–phonon matrix elements (which is necessary due to the diverging density of states close at the magic angle). This demonstrates that it is the off-diagonal (or so-called gauge phonon contributions[40,49,50]) of the acoustic phonon matrix element that dominates the transport properties. How to screen the electron–phonon matrix element in two dimensions has long remained controversial. The issue is that prior to the present work, calculating the full dynamical polarizability at finite temperature has been challenging. Since the phonon propagator couples at a particular frequency, without dynamical screening, it is unclear how the electrons screen the deformation potential. This led to speculation in the theoretical literature as to whether the deformation potential should be screened or left unscreened. For example, in ref. [51], Okuyuma and Tokuda argue that experimental data for GaAs 2DEGs is better fit using the unscreened theory, while in a later work, Kawamura and Das Sarma argue that once correctly done, static screening gives excellent agreement with experimental data[52]. Even for monolayer graphene, ref. [50] argue that the deformation potential is completely screened, while the ref. [53] argue for no screening. In this work, we demonstrate conclusively that as anticipated by Kawamura and Das Sarma, the static screening approximation is closer to the correct dynamically screened result than the commonly used unscreened approximation. The Supplementary Information has details of the screening as well a discussion of the geometric enhancement due to the moiré that is present for gauge phonons, but not scalar phonons.

**Interplay between intraband and interband scattering**. There are two qualitatively distinct regimes depending on whether the

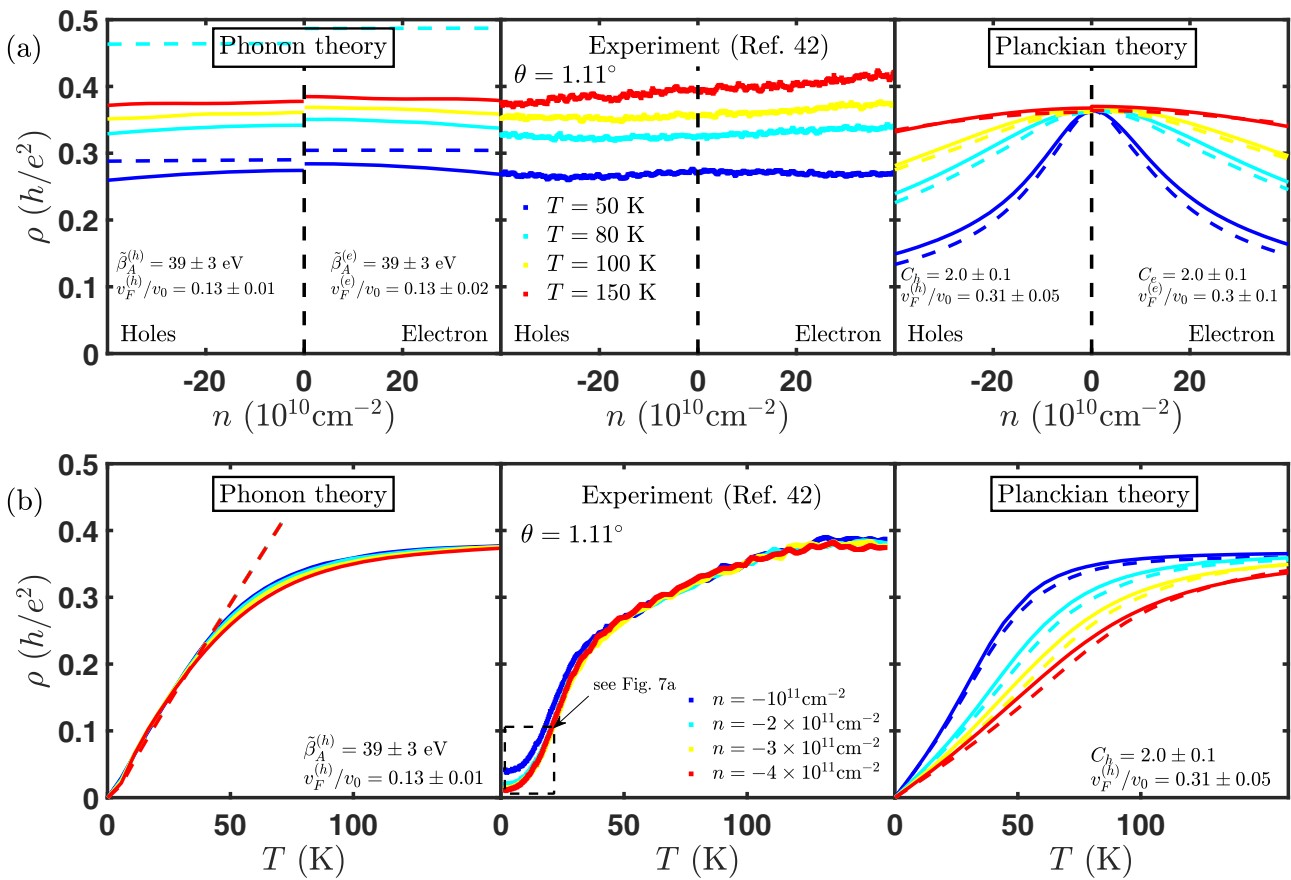

**Fig. 2 Experimental observations are consistent with the electron-phonon scattering theory.** The electron–phonon scattering theory (left panels) correctly captures the **a** carrier density and **b** temperature dependence of experimentally observed resistivity (middle panels), unlike the Planckian theory (right panel) that shows a stronger density dependence. Experimental data is taken from ref. [42] for $\theta = 1.11°$ (comparison for devices with other twist angles is shown in the Supplementary Information). Solid lines in the electron–phonon and Planckian theory are for a two-band effective model that includes the van Hove singularity, while the dashed lines are for the linear Dirac model. For electron–phonon scattering, the linear-in-$T$ resistivity at low temperature is captured by the Dirac model, while the saturation at higher temperature requires the van Hove singularity. For the Planckian theory, the Dirac model and the two-band model are quantitatively similar and show much stronger density dependence compared to the experiment. In this case, the saturation at high temperature is set not by the van Hove singularity, but by a universal value $\rho(T \to \infty) = C/(8\ln 2)h/e^2$ (the coefficient $C \leq 1$ for Planckian dissipation). For most experimental data, including those shown here, $C \geq 1$. Taken together with the weak density dependence seen experimentally, this suggests that phonon scattering rather than Planckian dissipation is the dominant scattering mechanism at play in twisted bilayer graphene.

phonon velocity $c_{ph}$ is greater than or smaller than the Fermi velocity $v_F$. The crossover from $v_F > c_{ph}$ to $v_F < c_{ph}$ is expected because the renormalized $v_F$ vanishes at magic angles $\theta_M$. For the largest magic angle, this crossover occurs at $\theta = \theta_{cr} \sim 1.15°$ and separates the regimes of interband ($v_F < c_{ph}$) and intraband ($v_F > c_{ph}$) scattering. Within the Dirac regime, the theory for intraband scattering is now well established[47,50,53–55]. The resistivity shows a Bloch–Grüneisen behavior similar to metals and is given by $\rho_{e-ph} = [16\zeta(\theta)^2 k_F/(e^2 \mu_s c_{ph} v_F^2)]F(T_{BG}/T)$, where $T_{BG} = 2\hbar c_{ph} k_F$ is the Bloch–Grüneisen temperature, which is the characteristic crossover scale over which the temperature dependence of the resistivity due to electron–phonon scattering changes from $T^4$ below $T_{BG}$ to $T$ − linear above $T_{BG}$. This change in the resistivity occurs due to the restricted scattering phase space of phonons at low temperatures when their quantum nature becomes important, compared to higher temperatures where the phonon distribution is quasiclassical. The graphene mass density is $\mu_s$, and $F(x) = \int_0^1 dy[xy^4\sqrt{1-y^2}e^{xy}]/(e^{xy}-1)^2$ (This form of the integral first appeared in ref. [53]). For $T \gg T_{BG}$, the quantization of the lattice phonon modes is irrelevant and the scattering is expected to be proportional to the amplitude of lattice

vibrations and is linear-in-T. In fact, one can show exactly that the linearity persists up to temperatures as low as $T_{BG}/4$.

The interband phonon scattering rate (dominant close to magic angle) shares some similarities with the intraband scattering: it is density-independent and $T$ − linear at high-$T$. Moreover, it vanishes when $v_F \to 0$ as the scattering phase space tends to zero. However, qualitatively the interband and intraband scattering are quite different (see Supplementary Information). For example, while the intraband scattering rate within the Dirac model shows a monotonic increase with energy, the interband scattering rate is non-monotonic highlighting the suppression of interband scattering for energies larger than $k_B T$. Most importantly, the temperature scale for the scattering rate to be $T$ − linear is not set by the Bloch–Grüneisen temperature $T_{BG}$, but rather by the Fermi temperature $T_F$ (which is the maximum phonon energy at the Fermi surface allowed by kinematic constraints). We note that $T_F$ and $T_{BG}$ are defined in such a way so that at $\theta_{cr}$, $T_F = T_{BG}/2$. Therefore, close to the magic angle when $v_F < c_{ph}$, we have $T_F < T_{BG}/2$ and the electron–phonon scattering becomes $T$ − linear for temperatures well below $T_{BG}$. The reversal of the temperature scales is due to the fact the $T_F$ depends on the twist angle dependence (through $v_F$). On the

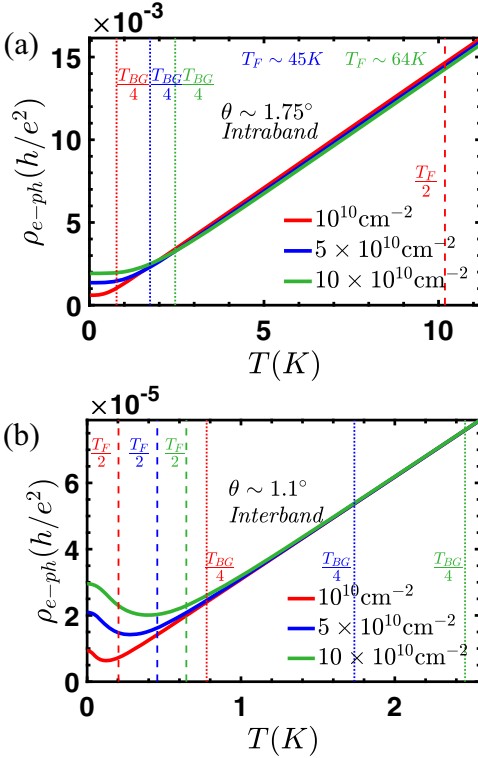

**Fig. 3 The persistence of linear-T behaviour to low temperature is due to inter-band scattering.** Electron–phonon resistivity for tBG within the Dirac model for **a** intraband scattering and **b** interband scattering. Dotted and dashed lines indicate $T_{BG}/4$ and $T_F/2$, respectively. Interband resistivity shows a transition to linear-in-$T$ at $\sim T_F/2$ compared to intraband resistivity which shows a transition around $\sim T_{BG}/4$. In the interband regime, the striking persistence of linear-in-$T$ behavior for $T \ll T_{BG}$ is observed.

other hand, $T_{BG}$ depends on phonon velocity and density, both of which do not change with twist angle. Thus at small twist angles, $T_F$ can become significantly smaller than $T_{BG}$. For $T \gg T_F$, we find

$$\rho_{\text{inter}}^{e-\text{ph}} = \frac{h}{e^2} \frac{2\tilde{\beta}_A^2 v_F^2 k_B T}{\hbar^2 \mu_s c_{\text{ph}}^6}, \tag{1}$$

where $\tilde{\beta}_A$ is the twist angle-dependent enhanced gauge field coupling constant. Close to magic angle, we expect $\rho_{\text{inter}}^{e-\text{ph}}(T \gg T_F) \propto v_F^4$ (where the additional $v_F^2$ comes from $\tilde{\beta}_A^2$, see Supplementary Information), and vanishes at the magic angle due to the lack of scattering phase space.

Figure 3a shows intraband resistivity for a chosen $\theta > \theta_{\text{cr}}$. The linear-in-$T$ resistivity persists down to $T_{BG}/4$ consistent with earlier results[47]. A comparison of the scales of $T_{BG}$ and $T_F$ is also done. Figure 3b shows the interband resistivity for a chosen $\theta < \theta_{\text{cr}}$, comparing the scales of $T_F$ and $T_{BG}$. The linearity in $T$ is observed to persist down to very low temperatures even when $T \ll T_{BG}$, which is very different from the known theory of electron–phonon scattering in a typical Fermi liquid. The empirical observation of $T$−linear resistivity well below $T_{BG}$ has been attributed to the strange metallicity of non-Fermi liquids[41], however, we find that there is nothing mysterious about this feature, it is merely the qualitative change in the nature of electron–phonon scattering when $v_F < c_{\text{ph}}$ i.e., the lower of the two energy scales switches from lattice vibrational energy to electronic energy when $v_F$ crosses below $c_{\text{ph}}$.

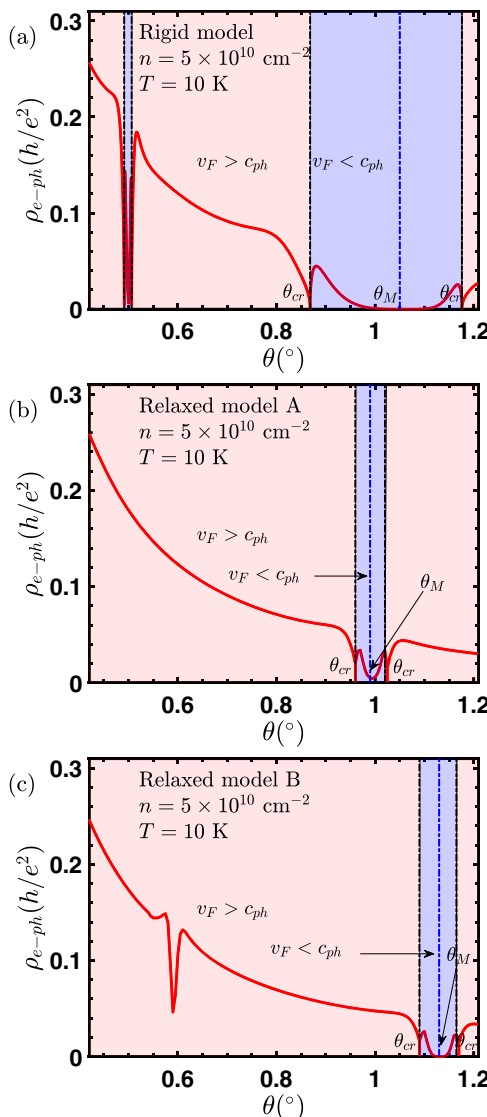

**Fig. 4 Close to the magic angle, the electron–phonon resistivity in tBG is very sensitive to the twist angle exhibiting a variation of several orders of magnitude when $v_F = c_{ph}$ and unrelated to Mott insulation or superconductivity.** The number of these sharp dips in resistivity and the angles at which they occur provide information about lattice relaxation[13]. **a** The rigid lattice model of Bistritzer and MacDonald[12] predicts that the resistivity will have multiple dips with decreasing twist angle (corresponding to three dips per magic angle). **b** The relaxation model of ref. [64] gives only three sharp dips close to a single value of the magic angle. **c** The relaxation model of ref. [58] gives three sharp dips close to a single value of the magic angle and another dip when $v_F$ becomes quite close to $c_{ph}$ but never goes below it.

**Lattice relaxation effects.** For small twist angles, atoms on both layers will tend to move away from their nominal positions in order to minimize the total energy of the system[13]. This relaxation process will increase the fraction of atoms with AB stacking relative to regions with AA stacking. This rearrangement of atoms also changes the relative strength of moiré coupling between different sublattices across the twisted interface. Relaxed atomic positions are calculated either using a continuum elasticity theory[56] for what we call "relaxed model A" or using the molecular dynamic approach as implemented in LAMMPS[57,58] that we call "relaxed model B". Once we know the relaxed atomic positions, the moiré coupling parameters are obtained by Fourier

transforming the matrix that couples the orbitals in layer 1 and layer 2. It is these "relaxed" hopping parameters that are then used in the continuum model Hamiltonian to calculate electronic bandstructure and corresponding renormalized Fermi velocity (see Fig. 4) that is used as the input for our Boltzmann transport calculation. While the two relaxation models have some quantitative differences such as the position of the magic angle, they both give the same qualitative description for the role of relaxation on the moiré bandstructure.

In Fig. 4 we plot the electron–phonon resistivity for both the rigid continuum model[12] of tBG as well as including the lattice relaxation effects using the two different models that we label as

wave vector separation between the two Dirac points in monolayer graphene and $\theta$ is twist angle. The two-band model is valid when the bandwidth is much larger than $\hbar v_F |\Delta K|$ which is a good approximation for small twist angles. The eigenenergies of this Hamiltonian are given by $\varepsilon_{\mathbf{k},\lambda} = \lambda(1/4)\hbar v_F \sqrt{k_\theta^2 + 8(k_x^2 - k_y^2) + 16[(k_x^2 + k_y^2)/k_\theta]^2}$, and are anisotropic. Within an isotropic approximation for the momentum-transport cross section, we calculate the electron–phonon scattering rate for both intraband ($v_F > c_{ph}$) and interband ($v_F < c_{ph}$) with only intravalley electron–phonon scattering (see Supplementary Information for details). We find

$$\frac{1}{\tau_{\text{intra}}^{e-\text{ph}}(r,\phi,\lambda)} = \sum_{\substack{\xi=\pm 1 \\ \nu=\text{TA,LA}}} \frac{\tilde{\beta}_A^2 k_\theta}{8\pi\mu_s\hbar c_\nu v_F} \int_0^\infty dr' r' \int_{-\pi}^\pi d\phi' \frac{q(r,\phi,r',\phi')\sin^2(\phi'-\phi)}{\sqrt{(1+r'\cos\phi')^2+(r'\sin\phi')^2}} \left((1-\xi)/2 + \xi f_{\mathbf{p}',\lambda}^0 + n_{\mathbf{q},\nu}\right)$$
$$\times \delta\left(\lambda(r'-r) - 4\xi z \frac{q(r,\phi,r',\phi')}{k_\theta}\right) \tag{3}$$

$$\frac{1}{\tau_{\text{inter}}^{e-\text{ph}}(r,\phi,\lambda)} = \sum_{\nu=\text{TA,LA}} \frac{\tilde{\beta}_A^2 k_\theta}{8\pi\mu_s\hbar c_\nu v_F} \int_0^\infty dr' \int_{-\pi}^\pi d\phi' \frac{q(r,\phi,r',\phi')\sin^2(\phi'-\phi)}{\sqrt{(1+r'\cos\phi')^2+(r'\sin\phi')^2}} \left((1+\lambda)/2 - \lambda f_{\mathbf{p}',-\lambda}^0 + n_{\mathbf{q},\nu}\right)$$
$$\times \delta\left(r'+r - 4z\frac{q(r,\phi,r',\phi')}{k_\theta}\right), \tag{4}$$

relaxed model A and relaxed model B. We first note that whenever $v_F = c_{ph}$, there are sharp dips in the resistivity profile, which can span a few orders of magnitude. Secondly, at each magic angle, there is another large dip. For the rigid lattice model, there is a broad window where $v_F < c_{ph}$ around the magic angle $\theta_M \sim 1.06°$, and the family of magic angles implies multiple crossings of $v_F = c_{ph}$, and hence multiple dips in the resistivity as the twist angle is lowered. With the inclusion of lattice relaxation effects, there is only a single stable magic angle (that is shifted slightly from the original magic angle), and there are therefore only three dips in the resistivity near each magic angle. The window of the interband scattering regime ($v_F < c_{ph}$) is also narrower. We find that for relaxed model B, near 0.6°, the Fermi velocity becomes quite close to $c_{ph}$ but never goes below it. This results in another sharp dip near that angle.

We emphasize that the following robust features survive with or without the inclusion of relaxation effects and are independent of the relaxation modeling: (i) existence of the critical angle $\theta_{cr}$ by construction, (ii) huge drop in resistivity at $\theta_{cr}$ and at $\theta_M$ resulting in a total of three sharp resistivity dips near each magic angle. Since close to magic angle, the resistivity is highly sensitive to twist angle exhibiting a variation of several orders of magnitude, one must be careful to experimentally distinguish this phonon effect from possible Mott insulation or superconductivity.

**Beyond the Dirac model.** While the Dirac model captures the physics of tBG at low density, in order to make accurate predictions for larger density and temperature, we need to extend the model to capture the VHS. We use an effective two-band Hamiltonian[48]

$$H(\mathbf{k}) = -\frac{\hbar v_F}{|\Delta \mathbf{K}|}\begin{pmatrix} 0 & k^{*2} - (\Delta \mathbf{K}^*/2)^2 \\ k^2 - (\Delta \mathbf{K}/2)^2 & 0 \end{pmatrix}, \tag{2}$$

where $v_F$ is the Fermi velocity of tBG at the Dirac point, $k = k_x + ik_y$, $\Delta \mathbf{K}$ is wave vector separation between the two Dirac points which are located at $\mathbf{K}$ and $\mathbf{K}_\theta$, which magnitude is given by $k_\theta \equiv |\Delta \mathbf{K}| = 2k_D \sin(\theta/2)$, with $k_D$ being the

where $z_\nu = c_\nu/v_F$, $q(r,\phi,r',\phi')$ is momentum transferred by phonon, and $f_{\mathbf{p}',\lambda}^0$ ($n_{\mathbf{q},\nu}$) is the equilibrium Fermi (Bose–Einstein) distribution function describing the electron (phonon) population. Here, $\xi = \pm 1$ refers to phonon absorption (+1) and emission (−1), respectively. The resistivity for this Hamiltonian is

$$\frac{1}{\rho_{ij}} = 8e^2 \sum_{\lambda=\pm 1} \int_0^\infty dr \int_{-\pi}^\pi d\phi \frac{\mathcal{J}(r,\phi)}{(2\pi)^2} v_{\mathbf{k},\lambda}^{(i)} v_{\mathbf{k},\lambda}^{(j)}$$
$$\times \tau^{e-\text{ph}}(r,\phi)\left(-\frac{\partial f_{\mathbf{k},\lambda}^0}{\partial \varepsilon_{\mathbf{k},\lambda}}\right) \tag{5}$$

where $v_{\mathbf{k},\lambda}^{(j)} = (1/\hbar)(\partial \varepsilon_{\mathbf{k},\lambda}/\partial k_j)$ is band velocity in $j$ direction and $\lambda = \pm 1$ refers to conduction (+1) and valence (−1) band, respectively (see Supplementary Information for the explicit form of $\mathcal{J}(r,\phi)$).

We plot the results of our calculation of $\langle \rho \rangle = \sqrt{\rho_{xx}\rho_{yy}}$ for twist angle of $\theta = 1.1°$ in the left panel of Fig. 2. The electron–phonon resistivity is linear-in-$T$ at low temperature but saturates at high temperature. The slope of resistivity with temperature in the low temperature regime is set by $v_F$ while the saturation of resistivity with temperature is set by the bandwidth. Within this effective Hamiltonian, the bandwidth $2\varepsilon_{\text{VHS}}$ and $v_F$ are not independent and related by $2\varepsilon_{\text{VHS}} = (1/2)\hbar v_F k_\theta$. While this relation between $\varepsilon_{\text{VHS}}$ and $v_F$ is specific to our model, we believe that the main conclusions are generic i.e., the low temperature linear-in-$T$ behavior is set by the Fermi velocity, while the saturation is set by the VHS. The saturation of electron–phonon resistivity can be simplified deep in the intraband ($v_F \gg c_{ph}$) and interband ($v_F \ll c_{ph}$) regime as

$$\rho_{\text{intra}}^{e-\text{ph}}(T \to \infty) \propto \frac{\tilde{\beta}_A^2}{e^2\hbar\mu_s}\frac{1}{v_F^2 c_{ph}^2}\varepsilon_{\text{VHS}} \tag{6}$$

$$\rho_{\text{inter}}^{e-\text{ph}}(T \to \infty) \propto \frac{\tilde{\beta}_A^2}{e^2\hbar\mu_s}\frac{v_F^2}{c_{ph}^6}\varepsilon_{\text{VHS}} \tag{7}$$

**Resistivity from the Planckian model.** We take a simple phenomenological model with $\hbar\tau_{\text{Pl}}^{-1} = C k_B T$. The resistivity is

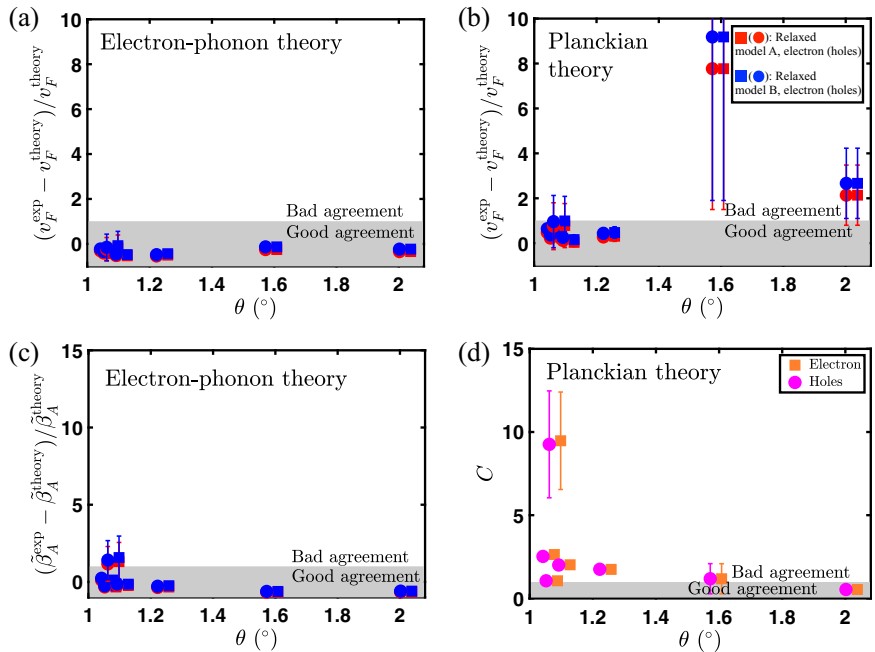

**Fig. 5 The parameters obtained from fit to refs. [2,42,62] (see Methods for details).** We find that the data agree much better with the expectations from the electron–phonon theory than the Planckian model. **a** Relative deviations of Fermi velocity as a function of twist angle obtained by fitting the experimental resistivity to the electron–phonon theory (**b**) obtained by fitting to the Planckian theory, **c** Relative deviations of the effective electron–phonon coupling constant $\tilde{\beta}_A$, and **d** Planckian strength $C$. The error bars represent 95% confidence interval of the parameters. The Planckian bound ($C \leq 1$) is violated for small twist angles rules out the Planckian model as the dominant transport mechanism in tBG.

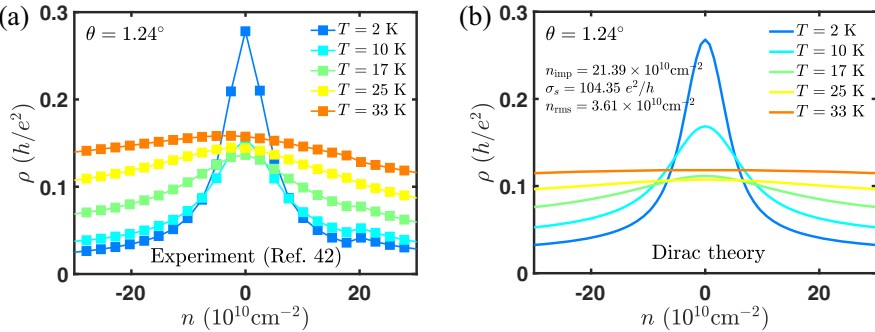

**Fig. 6 The weak density dependence at high temperature and strong density dependence at a low temperature of the resistivity close to charge neutrality seen in experiment[42] is well captured within the Dirac theory of electron-(gauge) phonon scattering[40].** Experimental data (top squares) for resistivity at $\theta = 1.24°$ vs density at various temperatures (**a**) compared to the Dirac theory (bottom solid lines) of electron–phonon scattering (**b**).

obtained from the Boltzmann equation by using the relaxation time approximation, with $\tau_{Pl}$ as the relaxation time (see discussion in ref. [43]), and calculating the appropriate thermal average with the density of states and Fermi velocity (see Eq. 19). We can do this both for the Dirac model and the two-band Hamiltonian. Surprisingly, both the Dirac Hamiltonian and the two-band model give very similar results indicating that the van Hove singularity is not important for the Planckian theory. This phenomenological model exhibits a linear-in-$T$ resistivity at low temperature that saturates at higher temperature (qualitatively similar to what is seen experimentally). We find that the slope of resistivity with temperature in the low temperature regime is set by both $\nu_F$ and $C$, while the saturation of resistivity with temperature is set only by $C$. Within the Dirac approximation, the resistivity is

$$\frac{1}{\rho_{Pl}} = \frac{1}{C}\frac{4e^2}{h}\sum_{\lambda=\pm 1}\ln\left[1 + \exp\left(\lambda\frac{\mu}{k_BT}\right)\right], \quad (8)$$

with low and high temperature asymptotes

$$\rho_{Pl} = \begin{cases} C\frac{h}{4e^2}\frac{T}{T_F}\left[1 + \frac{\pi^2}{6}\left(\frac{T}{T_F}\right)^2\right]; & T \ll T_F \\ C\frac{h}{e^2}\frac{1}{8\ln 2}\left[1 - \frac{1}{128(\ln 2)^3}\left(\frac{T_F}{T}\right)^4\right]; & T \gg T_F \end{cases} \quad (9)$$

For the effective two-band model, the Planckian resistivity is given by an expression similar to Eq. 5. It can be simplified to

$$\frac{1}{\rho_{Pl}^{ij}} = \frac{e^2}{h}\frac{1}{C}K_j\left(\frac{n}{n_{VHS}}, \frac{k_BT}{\varepsilon_{VHS}}\right), \quad (10)$$

where the function $K_j$ is computed numerically (see Supplementary Information). In the right panel of Fig. 2a, b, we show the Planckian resistivity as a function of density and temperature, respectively at twist angle of 1.11°. The Planckian resistivity for both models saturate at $\rho_{Pl}(T \to \infty) = C/8\ln 2$, independent of the bandwidth $\varepsilon_{VHS}$, which is in sharp contrast to electron–phonon scattering. Figure 5 shows the results of fitting

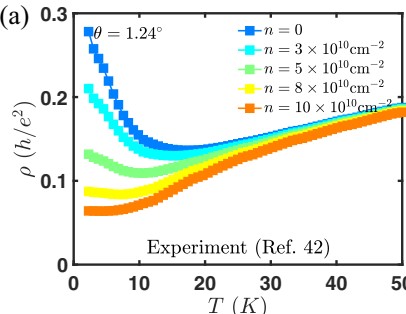
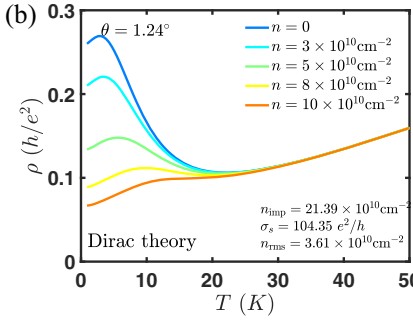

**Fig. 7 The non-monotonic temperature dependence at low temperatures and linear-in-$T$ dependence at intermediate temperatures seen in experiment is well captured within the Dirac theory of electron–(gauge) phonon scattering[40].** Resistivity vs temperature from **a** ref. [42] and **b** Dirac theory[40] at twist angle of $\theta = 1.24°$. The non-monotonicity at low temperatures is due to the crossover from impurity-dominated transport to phonon-dominated transport.

eight data sets of varying twist angle, temperature, and carrier density to both the phonon and Planckian models (full fits are shown in Supplementary Information). We find that the phonon scattering theory (but not the Planckian model) gives fit parameters for both the Fermi velocity and model parameters that are consistent with theoretical expectations.

The nature of electron–phonon scattering investigated here gives rise to several new features: Interband electron–phonon scattering, which should be kinematically forbidden in monolayer graphene, is shown to occur in tBG below a critical twist angle $\theta_{cr}$, when $v_F < c_{ph}$. The critical angle ($\theta_{cr}$) is a sweet spot where both the interband and intraband scattering phase space (and thus resistivity) both drop to zero, giving rise to multiple dips in the resistivity unrelated to insulating or superconducting states that could be investigated in the future experiments. We derive explicit analytical expressions for the interband scattering rate and show its qualitative dissimilarity from the intraband scattering. Importantly, we show that this explains the linear-in-$T$ resistivity well below the Bloch–Gruneisen temperature $T_{BG}$, a previously unexplained experimental puzzle.

In this work, we also provide additional theoretical verification of our earlier claim[40] that the gauge phonon modes are enhanced by the moiré geometry and are not screened, while the scalar phonon modes have neither property. We also provide experimental verification (see Figs. 6, 7) of our previous predictions that charged impurity scattering takes over as the dominant scattering mechanism at very low temperatures (below 20 K) and low carrier densities (below ~$10^{11}$cm$^{-2}$). Taken together with the present work on phonon scattering (that applies at high temperature and high carrier density), this now presents a complete theory for the carrier transport for tBG in the metallic regime.

## Methods

**Dynamical screening of electron–phonon coupling.** The basic building block of RPA screening is the polarizability bubble $\Pi_C(\mathbf{q}, \omega)$, which can be written in the most general form as

$$\Pi_C(\mathbf{q}, i\omega_m) = \frac{1}{\beta} \frac{g}{A} \sum_{\mathbf{k}, i\omega_n} \text{Tr}[G(\mathbf{k}, i\omega_n) G(\mathbf{k} + \mathbf{q}, i\omega_n + i\omega_m)], \quad (11)$$

where the summation $i\omega_n$ is over the imaginary frequency, $G(\mathbf{k}, i\omega)$ is the Green's function, and the trace is over the sublattice degrees of freedom. Performing the trace and the Matsubara summation we obtain

$$\Pi_C(\mathbf{q}, i\omega) = \lim_{\eta \to 0^+} \frac{g}{A} \sum_{\lambda, \lambda', \mathbf{k}} \frac{f^0_{\mathbf{k}, \lambda} - f^0_{\mathbf{k}+\mathbf{q}, \lambda'}}{\hbar\omega + \varepsilon_{\mathbf{k}, \lambda} - \varepsilon_{\mathbf{k}+\mathbf{q}, \lambda'} + i\eta} F^{\lambda\lambda'}_{\mathbf{k}, \mathbf{k}+\mathbf{q}}, \quad (12)$$

where $F^{\lambda\lambda'}_{\mathbf{k}, \mathbf{k}+\mathbf{q}}$ is the tBG chirality factor, which for Dirac model is given by $(1 + \lambda\lambda' \cos\theta_{\mathbf{k}, \mathbf{k}+\mathbf{q}})/2$ and $f^0_{\mathbf{k}, \lambda} = [\exp\{(\varepsilon_{\mathbf{k}, \lambda} - \mu)/(k_B T)\} + 1]^{-1}$ is the equilibrium Fermi distribution function. The dielectric function $\epsilon(\mathbf{q}, i\omega_m)$ is given by the RPA summation and is related to the basic pair bubble as $\epsilon(\mathbf{q}, \omega) = 1 - V_{\mathbf{q}} \Pi_C(\mathbf{q}, i\omega)$, where $V_{\mathbf{q}} = 2\pi e^2/\kappa q$ is the Fourier transform of the Coulomb potential, $\kappa$ being the dielectric constant. We evaluate $\Pi_C$ and $\epsilon(\mathbf{q}, \omega)$ for finite frequencies and

temperatures, semi-analytically. To the best of our knowledge, this has not been done previously in the literature, at least in the context of phonons (see Supplementary Information).

**Boltzmann transport.** The distribution function $f_{\mathbf{k}, \lambda}$ is evaluated within the Boltzmann transport formalism, which reads

$$-\frac{e\mathbf{E}}{\hbar} \nabla_{\mathbf{k}} f_{\mathbf{k}, \lambda} = \text{St}[f_{\mathbf{k}, \lambda}] \quad (13)$$

The collision integral can be written as

$$\text{St}[f_{\mathbf{k}, \lambda}] = \sum_{\mathbf{k}', \lambda', \nu} P^{\lambda'\lambda}_{\mathbf{k}'\mathbf{k}, \nu} f_{\mathbf{k}', \lambda'}(1 - f_{\mathbf{k}, \lambda}) - P^{\lambda\lambda'}_{\mathbf{k}\mathbf{k}', \nu} f_{\mathbf{k}, \lambda}(1 - f_{\mathbf{k}', \lambda'}), \quad (14)$$

where $P^{\lambda'\lambda}_{\mathbf{k}'\mathbf{k}, \nu}$ is the scattering probability from state $|\lambda', \mathbf{k}'\rangle$ to $|\lambda, \mathbf{k}\rangle$ within phonon branch $\nu$ (TA or LA), which is given by

$$P^{\lambda\lambda'}_{\mathbf{k}, \mathbf{k}+\mathbf{q}, \nu} = \frac{2\pi}{\hbar} \left|g^{\lambda\lambda'}_{\mathbf{k}, \mathbf{k}+\mathbf{q}, \nu}\right|^2 \left[n_{\mathbf{q}, \nu}\delta(\varepsilon_{\mathbf{k}+\mathbf{q}, \lambda'} - \varepsilon_{\mathbf{k}, \lambda} - \hbar\omega_{\mathbf{q}, \nu}) + (1 + n_{\mathbf{q}, \nu})\delta(\varepsilon_{\mathbf{k}+\mathbf{q}, \lambda'} - \varepsilon_{\mathbf{k}, \lambda} + \hbar\omega_{\mathbf{q}, \nu})\right], \quad (15)$$

where the two terms account for absorption and emission of phonons, $n_{\mathbf{q}, \nu}$ is the Bose–Einstein distribution function describing the phonon population, $\hbar\omega_{\mathbf{q}, \nu} = \hbar c_\nu q$ is the phonon energy, and $g^{\lambda\lambda'}_{\mathbf{k}, \mathbf{k}+\mathbf{q}, \nu}$ is the electron–phonon coupling, which can be expressed as

$$g^{\lambda\lambda'}_{\mathbf{k}, \mathbf{k}+\mathbf{q}, \nu} = \sqrt{\frac{\hbar}{2A\mu_s \omega_{\mathbf{q}, \nu}}} M^{\lambda\lambda'}_{\mathbf{k}, \mathbf{k}+\mathbf{q}} \quad (16)$$

where $A$ is the area of the graphene layer, $\mu_s$ is the mass density, and $M^{\lambda\lambda'}_{\mathbf{k}, \mathbf{k}+\mathbf{q}}$ is the matrix element for scattering between initial and final states, which is given by

$$M^{\lambda\lambda'}_{\mathbf{k}, \mathbf{k}+\mathbf{q}} = \zeta q \left[F^{\lambda\lambda'}_{\mathbf{k}, \mathbf{k}+\mathbf{q}}\right]^{1/2}, \quad (17)$$

where $\zeta$ is the effective deformation potential and $F^{\lambda\lambda'}_{\mathbf{k}, \mathbf{k}+\mathbf{q}}$ is the tBG chirality factor. The effective deformation potential $\zeta$ stands for either the effective scalar potential $\tilde{D}_A$ or twice the effective gauge potential $2\tilde{\beta}_A$ (see Supplementary Information).

We use ansatz

$$f_{\mathbf{k}, \lambda} = f^0_{\mathbf{k}, \lambda} + neE\tau_{\mathbf{k}, \lambda} \cos\theta_{\mathbf{k}} v_F \frac{\partial f^0_{\mathbf{k}, \lambda}}{\partial \varepsilon_{\mathbf{k}, \lambda}} \quad (18)$$

to obtain the transport scattering time $\tau_{\mathbf{k}, \lambda}$.

Finally the resistivity $\rho_{e-ph}$ is obtained from the scattering time $\tau^{e-ph}_{\mathbf{k}, \lambda}$ by

$$\frac{1}{\rho^{ij}_{e-ph}} = e^2 \frac{g^*}{A} \sum_{\mathbf{k}, \lambda = \pm 1} v^{(i)}_{\mathbf{k}, \lambda} v^{(j)}_{\mathbf{k}, \lambda} \tau^{e-ph}_{\mathbf{k}, \lambda} \left(-\frac{\partial f^0_{\mathbf{k}, \lambda}}{\partial \varepsilon_{\mathbf{k}, \lambda}}\right), \quad (19)$$

where $g^* = g$ for Dirac model and $g^* = g/2$ for two-band effective model ($g = 8$ is degeneracy), $v^{(j)}_{\mathbf{k}, \lambda} = (1/\hbar)(\partial \varepsilon_{\mathbf{k}, \lambda}/\partial k_j)$ is band velocity in $j$ direction, and $\lambda = \pm 1$ refers to conduction (+1) and valence (−1) band, respectively.

**Resistivity within the Dirac model.** In Fig. 6, we show the comparison of experimental resistivity vs density from ref. [42] to theoretical resistivity due to electron–phonon and electron-impurity scattering within Dirac model[40]. At low temperatures, electron–impurities dominates over electron–phonon interaction[40], therefore we only fit the experiment to the electron-impurity limited resistivity. These charged impurities also give rise to carrier density inhomogeneities[59], which cures the

**Table 1 Fitting parameters for the resistivity fit to the electron–phonon and Planckian theory. The parameters are Fermi velocity ($v_F$) and electron–phonon gauge coupling ($\tilde{\beta}_A$) for electron–phonon theory and Fermi velocity ($v_F$) and Planckian strength (C) for Planckian theory.**

| Twist angle | Electron–phonon | | Planckian | | Ref |
|---|---|---|---|---|---|
| | $v_F^{e(h)}/v_0$ | $\tilde{\beta}_A^{e(h)}$ (eV) | $v_F^{e(h)}/v_0$ | $C^{e(h)}$ | |
| 1.06° | 0.14 ± 0.01(0.17 ± 0.01) | 47 ± 2(51 ± 3) | 0.30 ± 0.03(0.35 ± 0.02) | 2.6 ± 0.1(2.5 ± 0.1) | 42 |
| 1.07° | 0.15 ± 0.01(0.15 ± 0.01) | 32 ± 1(32 ± 1) | 0.30 ± 0.01(0.30 ± 0.01) | 1.06 ± 0.03(1.06 ± 0.03) | 62 |
| 1.08° | 0.2 ± 0.1(0.2 ± 0.1) | 110 ± 60(100 ± 60) | 0.5 ± 0.2(0.4 ± 0.3) | 9 ± 3(9 ± 3) | 2 |
| 1.11° | 0.13 ± 0.02(0.13 ± 0.01) | 39 ± 3(39 ± 3) | 0.3 ± 0.1(0.31 ± 0.05) | 2.0 ± 0.1(2.0 ± 0.1) | 42 |
| 1.24° | 0.18 ± 0.03(0.17 ± 0.03) | 41 ± 4(39 ± 4) | 0.5 ± 0.1(0.5 ± 0.1) | 1.7 ± 0.1(1.7 ± 0.1) | 42 |
| 1.59° | 0.426 ± 0.001(0.426 ± 0.001) | 24.6 ± 0.1(24.6 ± 0.1) | 5 ± 4(5 ± 4) | 1 ± 1(1 ± 1) | 42 |
| 2.02° | 0.491 ± 0.002(0.491 ± 0.002) | 26.6 ± 0.2(26.6 ± 0.2) | 2 ± 1(2 ± 1) | 0.5 ± 0.3(0.5 ± 0.3) | 42 |

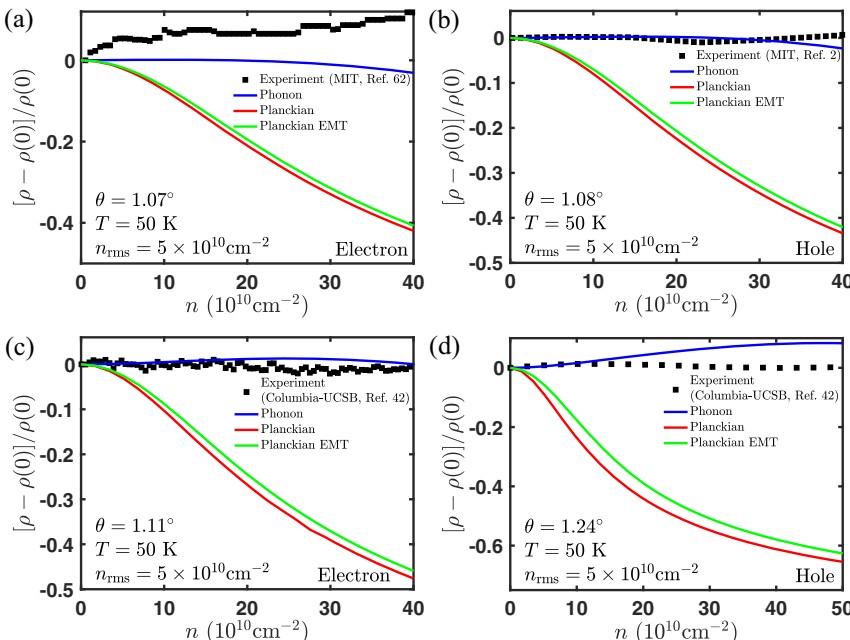

**Fig. 8 Alternative method for comparing the phonon theory (blue), Planckian theory (red), and experimental data (black) focusing only on the change in resistivity with carrier density.** We take four representative experimental curves from the full data set used to make Fig. 5, i.e., **a** $\theta = 1.07°$ (ref. [62]), **b** $\theta = 1.08°$ (ref. [2]), **c** $\theta = 1.11°$ (ref. [42]), and **d** $\theta = 1.24°$ (ref. [42]). Theory curves are the same as before, except we use the best theoretical estimates for $\beta_A$ and C. Also included is the Planckian theory with effective medium theory to include the effect of disorder smearing (green). Similar to the conclusions in Fig. 5, this method shows that the phonon theory has better agreement with the experiment.

otherwise divergent resistivity at the charge neutrality. We fixed the parameters from the effective medium theory (EMT)[60] fit to experimental resistivity at the lowest temperature, i.e., 2 K. In the fit, we have included a short-range scattering component of the conductivity $\sigma_s$[61], charged impurity density $n_{imp}$, and charge density fluctuations $n_{rms}$. The first two parameters are used to calculate the Boltzmann-RPA conductivity $\sigma_B[\sigma_s, n_{imp}]$, which takes into account both the dominant scattering mechanism of screened Coulomb impurities and additional scattering mechanisms due to short-range scatterers, such as point defects and line defects. The parameter $n_{rms}$ enters through the EMT equations. We find that the 1.24° device of ref. [42] has charged impurity density $n_{imp} = 2.1 \times 10^{11} \text{cm}^{-2}$, short-ranged conductivity $\sigma_s = 104 e^2/h$ and charge density fluctuations $n_{rms} = 3.6 \times 10^{10} \text{cm}^{-2}$. At a temperature of 10 K and density of $\sim 10^{11} \text{cm}^{-2}$, electron–phonon interaction start to produce a noticeable influence on the resistivity. We observe the reversal of temperature dependence trend at 10 K that is followed by linear-in-$T$ dependence at intermediate temperatures (see Fig. 7) and weaker density dependence at higher temperatures in an experiment. These were all predictions we made in our previous work[40] that have been now shown experimentally to be correct.

**Fitting of experimental data.** In Fig. 2, we compare the electron–phonon resistivity $\langle \rho \rangle = \sqrt{\rho_{xx}\rho_{yy}}$ from the two-band effective model (left panel) with the experiment from ref. [2,42,62] (middle panel) at a twist angle of $\theta = 1.1°$. The fitting parameters are Fermi velocity $v_F$ and the enhanced gauge field coupling constant $\tilde{\beta}_A$, which are obtained from fitting the temperature dependence of resistivity at density $n = 10^{11} \text{cm}^{-2}$ for a fixed twist angle. The dependence of the relative deviations of $v_F$

and $\tilde{\beta}_A$ on twist angle are plotted in Fig. 5a, c, respectively. The same parameters are used to plot the temperature dependence of electron–phonon resistivity at higher density as well as its density dependence at several fixed temperatures (see the left panel of Fig. 2b, a, respectively). We fit the electron-side and hole side separately due to a slight asymmetry between them. We find that in the low temperature regime, our results coincide with the Dirac model as expected, since the Hamiltonian in Eq. 2 reduces to a Dirac Hamiltonian at low energies. However, in the high temperature regime, only two band effective model agrees with the experiment (see Fig. 2). We see a clear saturation of the resistivity at higher temperatures as well as a weak density dependence in the electron and the hole side, as seen in the experiment. Both of these features are not captured within the Dirac model. From the fitting of the electron–phonon theory to the experiment (see Fig. 5c), we obtain $\beta_A$ around $1 - 9$ eV, which is in good agreement with the accepted values for the gauge field coupling constant in monolayer graphene[50,63]. The fit values for the velocity ratio $v_F/v_0$ also agree well with theoretical estimates[12]. Hence, we have developed a transport theory of tBG which explains all the salient features observed in the entire metallic regime (intermediate and high temperatures) of the experiment[42].

**Comparison with Planckian Resistivity.** We also compared the Planckian resistivity to the same experiment in Fig. 2 (right panel and middle panel, respectively). The fitting parameters are Fermi velocity $v_F$ and Planckian strength C. The fitting parameters of the resistivity fit to both the electron–phonon and Planckian theory for all the data sets are shown in Table 1. These are then used to plot the temperature dependence of Planckian resistivity at higher densities, as well as its

dependence on density at several fixed temperatures (see the right panel of Fig. 2b, a, respectively). Although the Planckian theory exhibits similar linear-in-$T$ behavior at low temperature and saturation at high temperature, its density dependence is much stronger than that of the experiment, especially at low temperatures (see Fig. 8). The Fermi velocity $v_F$ extracted from the fit is somewhat larger than the theoretical prediction (see Fig. 5). Moreover, we observe that the Planckian bound ($C \leq 1$) is violated for a small twist angle (see Fig. 5c). This rules out the Planckian theory as the dominant mechanism of metallic transport in tBG. We illustrate this point in Fig. 5 where we compare the relative deviations between theory and experiment. In order to compare the various theories and experiment on equal footing, these deviations were determined by shifting the magic angle for the theoretical Fermi velocity such that they all have a common magic angle that minimized the relative deviations of the parameters from their theoretical expectations—the x-axis could be therefore be interpreted as twist angle with the magic angle taken to be $\theta_M = 0.72°$.

## Data availability

The data used to generate the figures are available at https://github.com/indrayudhistira/tBG_metallic_phonon.

## Code availability

The code used to generate the figures are also available at https://github.com/indrayudhistira/tBG_metallic_phonon.

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

## Acknowledgements

We acknowledge the Singapore Ministry of Education AcRF Tier 2 (Grant No. MOE2017-T2-2-140), the Singapore National Research Foundation Investigator Award (Grant No. NRF-NRFI06-2020-0003), and the use of the dedicated research computing resources at CA2DM. M.S.F. and S.A. acknowledge the support of the ARC through grants CE170100039 and DP200101345. It is a pleasure to thank C. Dean, E. Laksono, P. Jarillo-Herrero H. Mahalingam, N. Raghuvanshi, and M. Yankowitz for fruitful discussions.

## Author contributions

G.S. and I.Y. contributed equally. S.A. supervised the project. G.S. calculated the dynamical screening of the electron–phonon coupling with assistance from D.Y.H.H. and G.V. I.Y. and N.C. calculated the electron–phonon resistivity beyond the Dirac model. I.Y. developed the Planckian model with assistance from M.S.F. G.S. and I.Y. calculated the electron–phonon resistivity with input from M.M.A.E. G.S., I.Y., N.C., and S.A. wrote the manuscript with input from all authors.

## Competing interests

The authors declare no competing interests.
