## [Peer Review File · Nature Communications]

Reviewers' Comments:

Reviewer #1:

Remarks to the Author:

In "Carrier transport theory for twisted bilayer graphene in the metallic regime" the authors present a theory for phonon-dominated transport in metallic twisted bilayer graphene, contrasting it with the alternative Planckian dissipation mechanism for resistivity.

The manuscript consists of the following parts:

1. A discussion of the state of the art concerning carrier transport in twisted BLG, both theoretically and experimentally. In particular the authors compare the experimental findings of Refs [40] and [13] and conclusions therein, arguing for a Planckian dissipation mechanism, or phonon-dominated transport, respectively.
2. The calculation of resistivity from electron-phonon scattering in multiple scenarios:
 - * in the Dirac regime for interband and intraband scattering
 - * Using an effective two band Hamiltonian that captures the van Hove singularity between the two Dirac points, again for interband and intraband scattering
3. The calculation of resistivity in a phenomenological Planckian model using the relaxation time approximation (again, for the Dirac regime and using the effective two-band model)
4. A comparison between the Electron-Phonon theory and the Planckian theory, as well as the comparison of both to previous experimental data. The authors argue that this comparison supports the case that transport in twisted bilayer graphene is phonon-dominated rather than of Planckian type.

The manuscript is well written, and the structure and logic are clear. The results are compelling and may advance the general understanding of transport in twisted bilayer graphene in the metallic regime. Therefore, in my opinion, the work can be considered for publication in Nature Communications. However, I have some concerns as to how lattice relaxation is included and discussed in this work (see points 2, 10, and 11 in the list below) and some further comments/questions to be addressed before publication:

1. In the abstract, the authors mention that it is "required to go well beyond the usual treatment" of electron-phonon scattering (and again on page 3, line 188). It would be instructive to explain (possibly somewhere in the text) why this is required and where precisely the "usual treatment" fails.
2. Page 2, line 99, the authors state that, since their special angle θ_{cr} is larger than the BLG magic angle, "its effects should be robust to lattice relaxation effects." This statement needs more explanation. What is the critical angle below which lattice reconstruction in BLG becomes significant (possibly, from the literature, such as arXiv:2007.01993 for twisted BLG or Phys. Rev. Lett. 124, 206101 (2020) for twisted TMDs)? The authors include lattice relaxation in their analysis of electronic structure later in the manuscript. Shouldn't it then be possible for them to test whether their results are robust against these effects, themselves?
3. Page 2, line 117, the authors state that it is "unclear if the non-interacting bands are stable to the 117 long-range Coulomb interaction". What are the implications for their own results and the statements of this manuscript?
4. Page 3. Figure 2: what is the reason for the resistivity offset at zero density in the left panel of the row a)?
5. Page 3, line 199: "Since phonons appear to dominate the transport properties, it is imperative that the theory is done correctly." I would encourage the authors to reconsider this sentence. It is

always important for any theory to be done correctly.

6. Page 4, Fig 3. The authors discuss the persistent linear-in-T behaviour for $T \ll T_{BG}$ for inter-band scattering, both in the figure caption and later in the text. However, also for intra-band scattering, panel a), linear behaviour seems to persist up to $T_{BG} / 4$, which is smaller than T_{BG} . Could the authors comment on that? Also, what is the physical origin of the reversed temperature scales for intra- and inter-band scattering ($T_F/2 > T_{BG}/4$ in panel a but $T_F/2 < T_{BG}/4$ in panel b)?

7. Page 5, line 282. Introducing the Bloch-Grueneisen temperature, T_{BG} , would merit a comment as to its origin and physical meaning.

8. Page 5, line 289, I got confused when the authors talk about "the intraband phonon scattering rate, dominant close to magic angle." Since v_F vanishes at the magic angle (as stated on page 4), shouldn't then be $v_F < c_{ph}$ and hence *inter*-band scattering be dominant near the magic angle (again, following the author's own logic on page 4)?

9. Page 5, line 315, discusses Figure 3b, but I wonder where the text discusses Figure 3a.

10. Page 5, line 330: The authors mention they include lattice relaxation effects. It needs some explanation as to how lattice relaxation is included, at what level, and using what technique, other than just citing ref 16. Also, related to my question 2: If the authors include lattice relaxation in their model, they should be able to test the stability of their results against lattice relaxation for different twist angles.

11. Page 5 Figure 4: In fact, there seems to be quite a change, both quantitatively and qualitatively, in the results for resistivity once lattice relaxation is included (again related to my comments 2 and 10). Most noticeably, the magic angle θ_M decreases from ~ 1.06 for the rigid lattice to 0.85 in the relaxed model. How does this go with the author's comment on page 2, line 98, that " θ_{cr} is larger than the original magic angle, and therefore its effects should be robust to lattice relaxation effects"? From Fig 4, I take that a) θ_{cr} in the relaxed model is, in fact, smaller than the original magic angle of ~ 1.06 , and b) gets modified significantly by lattice relaxation. Hence, reconstruction should, indeed play an important role. The authors need to rectify these inconsistencies.

12. Page 6, line 389, the authors mention using the relaxation time approximation in the Boltzmann equation. How well is the RTA justified in this case? For example, moving away from the Dirac points to the van Hove singularity, the dispersion is anisotropic, leading to tensor-like relaxation time. Or, what about possible density dependence of the relaxation time?

13. Page 7 Figure 5: The authors try to demonstrate that the parameters obtained from a fit to pre-existing experimental results agree better with the electron-phonon theory than with the Planckian model. Not all the cases portrayed are convincing. In panel c), the experimental parameters seem to show the opposite dependence on the twist angle (decreasing) that the theoretical curves (increasing). Panel b) is hard to read due to the split and the different scales in the upper and lower plot. Moreover, the lower figure's experimental results do not seem to disagree too much from the theoretical results.

I would strongly recommend revising this figure. One suggestion would be to plot relative deviations rather than absolute values to demonstrate agreement/disagreement across different scales. Moreover, it does not become clear from the main text why there is one panel showing data for "rigid (no geometry enhancement)" and to which calculation this curve corresponds.

14. As a general comment. At length, the paper discusses and even reuses the results of ref 13, of which they argue in favour. Conversely, the results of ref 40, which they claim to disprove, are mentioned, but not discussed in detail. The manuscript would greatly benefit from a more balanced discussion of both sides.

Reviewer #2:

Remarks to the Author:

In the present manuscript Sharma et. al. address the transport phenomena (resistivity) in twisted bilayer graphene (TBLG) near the magic angle. They mostly considered the phononic contributions and compared their findings with experiments, and also contrasted them with the Planckian theory (Ref. 40). By solving the Boltzman transport equation at finite temperatures, the authors arrive at the conclusion that phonon driven transport phenomena is more consistent with the existing experimental observations, while the existing Planckian theory shows some discrepancies. The subject of this article is interesting and definitely of current interest due to the ongoing experimental works on TBLG near the magic angle. As far as I understand, the main punchline of this work is the suitability of phonon driven transport in TBLG near the magic angle to match with existing experiments, which otherwise cannot be reconciled by the Planckian mechanism. Therefore, I will judge the suitability of this article for the publication in Nature Communication based on this point. On this ground I bear reservation against the publication of this work (at least in the current format) for the following reasons.

1. It is obvious from Fig. 2(b) that the Planckian theory is more consistent with experiments than the phonon theory. Specifically at high temperatures Planckian theory provides saturation of resistivity, which a pure phonon theory fails to capture.
2. Clearly, experimental data from Fig. 8 is not symmetric about zero doping. But phonon theory as well as Planckian theory produce symmetric curves for the resistivity.
3. Also the phonon driven Boltzman transport has already been addressed in great details in PRB 93, 195103 (2016). Authors should definitely cite this work. In that regard, I do not see any novelty in the calculation, besides extending the existing analysis beyond the Dirac picture. Also it is not obvious from any figure, how important are the higher order in momentum terms (beyond Dirac picture). In order words, authors need to explicitly show the discrepancy between the results obtained from pure Dirac theory and beyond Dirac theory.
4. Also from Fig. 9, it is obvious that Planckian theory yields better agreement with experiment at least at low temperatures. Therefore, the present theory fails to answer a simple question: Why phonon mediated transport then the absolute theory?

If the authors can provide convincing and quantitative arguments against all these issues or questions, I will reconsider the revised manuscript for publication in Nature Communication.

Also the authors should address the following (somewhat minor comments).

1. Authors say that a drop of resistivity by 5 orders is considered as a sign of superconductivity. They need to provide references for this statement. Presently it appears somewhat ad hoc.
2. Authors say that presently it is not clear whether the insulator is Mott insulator or Wigner crystal. The insulator can be a Slater insulator. Authors should mention it clearly.
3. Authors say that the role of the long range Coulomb interaction is not clear in TBLG. Note that TBLG is well described by slow Dirac fermions. For Dirac fermions it has been clearly established that long range Coulomb interactions can only lead to logarithmic increase of the Fermi velocity. See PRL118, 026403 (2017).
4. Authors say Planckian theory in explaining experimental data from Ref. 13 violates the assumption for the validity of the Planckian theory. How?
5. Authors claim that phonons "appear" to dominant transport. But, it is not at all obvious from any figure. One can easily conclude that Planckian theory is equally good or bad.
6. Authors also report multiple (2 or 3) dips in the resistivity. But, I could not see any dip in resistivity in any experimental data.

Therefore, in the current format the manuscript contains inconclusive and sometimes inconsistent (with experiment) results. Presently, it stands as a good technical paper, which is more suited for specialized journals (like PRB). Therefore, convincing quantitative arguments are needed, since the main claim of the paper is the consistency with experimental data.

Remarks of Referee 1:

Referee Remarks: The manuscript is well written, and the structure and logic are clear. The results are compelling and may advance the general understanding of transport in twisted bilayer graphene in the metallic regime. Therefore, in my opinion, the work can be considered for publication in Nature Communications. However, I have some concerns as to how lattice relaxation is included and discussed in this work (see points 2, 10, and 11 in the list below) and some further comments/questions to be addressed before publication:

Author Response: We thank the referee for the praising our work. We have addressed the lattice relaxation issue and all of the referee's other comments/questions below.

Referee Comment 1: In the abstract, the authors mention that it is "required to go well beyond the usual treatment" of electron-phonon scattering (and again on page 3, line 188). It would be instructive to explain (possibly somewhere in the text) why this is required and where precisely the "usual treatment" fails.

Author Response 1: In the revised manuscript we have now explained why this is required and where the usual treatment fails by adding following lines in the revised manuscript:

Line 218-230 of the revised manuscript: The usual treatment of the electron-phonon does not take into account interband scattering, that we find below to be crucial near the magic angle. Secondly, dynamical screening of phonons is completely neglected because typically $v_F \gg c_{ph}$, which again no longer holds true near the magic angle. Thirdly, geometric enhancement of the the gauge phonon mode remains poorly addressed, which we find below to be the most dominant phonon mode in tBG. Lastly, the usual treatment of the electron-phonon problem has a limited validity and fails beyond the linear regime (near VHS) that pushes us to go beyond the Dirac approximation and include non-linear lattice effects.

Referee Comment 2: Page 2, line 99, the authors state that, since their special angle θ_{cr} is larger than the BLG magic angle, "its effects should be robust to lattice relaxation effects." This statement needs more explanation. What is the critical angle below which lattice reconstruction in BLG becomes significant (possibly, from the literature, such as arXiv:2007.01993 for twisted BLG or Phys. Rev. Lett. 124, 206101 (2020) for twisted TMDs)? The authors include lattice relaxation in their analysis of electronic structure later in the manuscript. Shouldn't it then be possible for them to test whether their results are robust against these effects, themselves?

In the revised manuscript, we have now also tested our results with two different relaxation models (one uses an atomistic a continuum elasticity theory, and the other uses molecular dynamics relaxation). We call these "relaxed model A" and "relaxed model B", respectively; and show that our results are robust to inclusion of relaxation effects. We find that the following features are robust and survive with or without the inclusion of relaxation effects and independent of the relaxation modeling: (i) existence of the critical angle θ_{cr} , (ii) huge drop in resistivity at θ_{cr} and at θ_M resulting in a total of three

sharp resistivity dips near each magic angle. To clarify our point we have now added the following content in our revised manuscript:

Line 97-99 of the new manuscript: By construction, each magic angle must be accompanied by two critical angles θ_{cr} (above and below θ_M).

Line 418-428 of the revised manuscript: We emphasize that close to magic angle, the resistivity is highly sensitive to twist angle exhibiting a variation of several orders of magnitude and one must be careful to experimentally distinguish this from Mott insulation or superconductivity. Further, we note also that the following robust features survive with or without the inclusion of relaxation effects and independent of the relaxation modeling: (i) existence of the critical angle θ_{cr} by construction, (ii) huge drop in resistivity at θ_{cr} and at θ_M resulting in a total of three sharp resistivity dips near each magic angle.

Line 373-396 of the new manuscript: For small twist angles, atoms on both layers will tend to move away from their nominal positions in order to minimize the total energy of the system [15]. This relaxation process will increase the fraction of atoms with AB stacking relative to regions with AA stacking. This rearrangement of atoms also changes the relative strength of moiré coupling between different sublattices across the twisted interface. Relaxed atomic positions are calculated either using a continuum elasticity theory [58] for what we call “relaxed model A” or using molecular dynamic approach as implemented in LAMMPS [55,59] that we call “relaxed model B”. Once we know the relaxed atomic positions, the moiré coupling parameters are obtained by Fourier transforming the matrix that couples the orbitals in layer 1 and layer 2. It is these “relaxed” hopping parameters that are then used in the continuum model Hamiltonian to calculate electronic band-structure and corresponding renormalized Fermi velocity (see Fig. 4) that is used as the input for our Boltzmann transport calculation. While the two relaxation models have some quantitative differences such as the position of the magic angle, they both give the same qualitative description for the role of relaxation on the moiré bandstructure.

Revised figure 4 that now compares resistivity as a function of the twist angle for rigid model, and the two relaxed models.

Referee Comment 3: Page 2, line 117, the authors state that it is “unclear if the non-interacting bands are stable to the 117 long-range Coulomb interaction”. What are the implications for their own results and the statements of this manuscript?

In the revised manuscript we have clarified this sentence as follows:

Line 117-121 of the revised manuscript: Furthermore, it is possible for long-range interactions to significantly distort the non-interacting bands away from charge neutrality due to the formation of inhomogenous electrostatic potentials (see e.g. Refs. 40 and 41) although the experiments seem to suggest otherwise.

We point out that if the bands were reconstructed completely by the long-range interactions then features like the non-interacting bandwidth would not be observable in experiments. Since these are observed, this indicates that the non-interacting band structure is

a good starting point to understand the experimental data.

Referee Comment 4: Page 3. Figure 2: what is the reason for the resistivity offset at zero density in the left panel of the row a)?

Author Response 4: The offset at zero density of Fig. 2 top panel is because we fit each of the electron and hole sides of the experimental resistivity data with different fitting parameters to account for the particle-hole asymmetry in near magic angle tBG. We have added this explanation and the relevant reference in the main text, i.e.

Line 202-209 of the revised manuscript: Near magic angle tBG has electron-hole asymmetry that arises from the second-nearest hopping in the effective tight-binding model for graphene. The superlattice potential renormalizes the kinetic energy scales including the asymmetry to lower energies [14]. We include this effect in the theory by fitting separately for electron and hole side (as seen by the red and blue points in Fig. 5)

Referee Comment 5: Page 3, line 199: “Since phonons appear to dominate the transport properties, it is imperative that the theory is done correctly.” I would encourage the authors to reconsider this sentence. It is always important for any theory to be done correctly.

Author Response 5: We have now rephrased this sentence in the revised manuscript as follows:

Line 212-214 of the revised manuscript: Since phonons appear to dominate the transport properties, we are motivated to carefully consider the role of electron-phonon scattering.

Referee Comment 6: Page 4, Fig 3. The authors discuss the persistent linear-in- T behaviour for $T \ll T_{\text{BG}}$ for inter-band scattering, both in the figure caption and later in the text. However, also for intra-band scattering, panel a), linear behaviour seems to persist up to $T_{\text{BG}}/4$, which is smaller than T_{BG} . Could the authors comment on that? Also, what is the physical origin of the reversed temperature scales for intra- and inter-band scattering ($T_{\text{F}}/2 > T_{\text{BG}}/4$ in panel a but $T_{\text{F}}/2 < T_{\text{BG}}/4$ in panel b)?

The linear-in- T behavior for intraband scattering persisting until $T_{\text{BG}}/4$ is consistent with earlier results (e.g. Ref. 47). The key point to be noted here is that for intraband scattering the linear-in- T behavior is set by the temperature scale T_{BG} (which signifies a crossover from small-angle to large angle scattering), but for interband scattering the linear-in- T behavior is set by T_{F} . The reversal of the temperature scales is due to the fact the T_{F} depends on the twist angle dependence (through v_{F}), while T_{BG} depends on phonon velocity and density, both of which don't change with twist angle. Therefore, at small twist angles, T_{F} can become significantly smaller than T_{BG} . To clarify this point we have added the following lines in the revised manuscript:

Line 322-324 of the revised manuscript: In fact, one can show exactly that the linearity persists up to temperatures as low as $T_{\text{BG}}/4$.

Line 357-358 of the revised manuscript: The linear-in- T resistivity persists down to $T_{\text{BG}}/4$

consistent with earlier results [47].

Line 344-350 of the revised manuscript: The reversal of the temperature scales is due to the fact the T_F depends on the twist angle dependence (through v_F). On the other hand T_{BG} depends on phonon velocity and density, both of which don't change with twist angle. Thus at small twist angles, T_F can become significantly smaller than T_{BG} .

Referee Comment 7: Page 5, line 282. Introducing the Bloch-Grueneisen temperature, T_{BG} , would merit a comment as to its origin and physical meaning.

Author Response 7: We have now introduced the Bloch-Grueneisen temperature, T_{BG} in the revised manuscript as follows:

Line 309-317 of the revised manuscript: $T_{BG} = 2\hbar c_{ph} k_F$ is the Bloch-Grüneisen temperature, which is the characteristic crossover scale over which the temperature dependence of the resistivity due to electron-phonon scattering changes from T^4 below T_{BG} to T -linear above T_{BG} . This change in the resistivity occurs due to the restricted scattering phase space of phonons at low temperatures when their quantum nature becomes important, compared to higher temperatures where the phonon distribution is quasiclassical

Referee Comment 8: Page 5, line 289, I got confused when the authors talk about "the intraband phonon scattering rate, dominant close to magic angle." Since v_F vanishes at the magic angle (as stated on page 4), shouldn't then be $v_F < c_{ph}$ and hence *inter*-band scattering be dominant near the magic angle (again, following the author's own logic on page 4)?

This was a typo. It has been corrected in the revised manuscript. We thank the Referee for pointing this out.

Line 325-328 of the revised manuscript: The interband phonon scattering rate (dominant close to magic angle) shares ...

Referee Comment 9: Page 5, line 315, discusses Figure 3b, but I wonder where the text discusses Figure 3a.

A discussion on Fig. 3a is now presented in the main manuscript as follows:

Line 356-359 of the revised manuscript: Fig. 3a shows intraband resistivity for a chosen $\theta > \theta_{cr}$. The linear-in- T resistivity persists down to $T_{BG}/4$ consistent with earlier results [47]. A comparison of the scales of T_{BG} and T_F is also done.

Referee Comment 10: Page 5, line 330: The authors mention they include lattice relaxation effects. It needs some explanation as to how lattice relaxation is included, at what level, and using what technique, other than just citing ref 16. Also, related to my question 2: If the authors include lattice relaxation in their model, they should be able to test the stability of their results against lattice relaxation for different twist angles.

In the revised manuscript we have now added a significant discussion on lattice re-

laxation models that we have adopted for our procedure as presented below. We also added a co-author M. M. Al Ezzi who computed the band structures from the Large-scale Atomic/Molecular Massively Parallel Simulator (LAMMPS) for us to include in this manuscript. Our key results are qualitatively robust to lattice relaxation effects. To answer the Referees comment we have added the following discussion in the revised manuscript:

Line 373-396 of the new manuscript: For small twist angles, atoms on both layers will tend to move away from their nominal positions in order to minimize the total energy of the system [15]. This relaxation process will increase the fraction of atoms with AB stacking relative to regions with AA stacking. This rearrangement of atoms also changes the relative strength of moiré coupling between different sublattices across the twisted interface. Relaxed atomic positions are calculated either using a continuum elasticity theory [58] for what we call “relaxed model A” or using molecular dynamic approach as implemented in LAMMPS [55,59] that we call “relaxed model B”. Once we know the relaxed atomic positions, the moiré coupling parameters are obtained by Fourier transforming the matrix that couples the orbitals in layer 1 and layer 2. It is these “relaxed” hopping parameters that are then used in the continuum model Hamiltonian to calculate electronic band-structure and corresponding renormalized Fermi velocity (see Fig. 4) that is used as the input for our Boltzmann transport calculation. While the two relaxation models have some quantitative differences such as the position of the magic angle, they both give the same qualitative description for the role of relaxation on the moiré bandstructure.

Line 418-428 of the revised manuscript: We emphasize that close to magic angle, the resistivity is highly sensitive to twist angle exhibiting a variation of several orders of magnitude and one must be careful to experimentally distinguish this from Mott insulation or superconductivity. Further, we note also that the following robust features survive with or without the inclusion of relaxation effects and independent of the relaxation modeling: (i) existence of the critical angle θ_{cr} by construction, (ii) huge drop in resistivity at θ_{cr} and at θ_M resulting in a total of three sharp resistivity dips near each magic angle.

Referee Comment 11: Page 5 Figure 4: In fact, there seems to be quite a change, both quantitatively and qualitatively, in the results for resistivity once lattice relaxation is included (again related to my comments 2 and 10). Most noticeably, the magic angle θ_M decreases from ~ 1.06 for the rigid lattice to 0.85 in the relaxed model. How does this go with the author’s comment on page 2, line 98, that “ θ_{cr} is larger than the original magic angle, and therefore its effects should be robust to lattice relaxation effects?” From Fig 4, I take that a) θ_{cr} in the relaxed model is, in fact, smaller than the original magic angle of ~ 1.06 , and b) gets modified significantly by lattice relaxation. Hence, reconstruction should, indeed play an important role. The authors need to rectify these inconsistencies.

We thank the Referee for this comment that allows us to re-emphasize and clarify our point. In the revised manuscript we have reworked on the relaxation model and also considered a new relaxation model. We dub these models as relaxed model A based on the continuum theory, and relaxed model B based molecular dynamic approach as imple-

mented in LAMMPS. By construction, each magic angle is accompanied by two critical angles (one above and one below the magic angle), which are defined as the twist angles when the phonon velocity becomes equal to the Fermi velocity. Since the existence of at least one magic angle is guaranteed even in the presence of lattice relaxation effects, this automatically guarantees the existence of critical angle as well robustness to lattice relaxation effects. We find that in both the relaxation models that we adopt the magic angle shifts slightly from the original magic angle and the interband scattering window (between the magic angle and the critical angles on either side) is reduced. However the essential predictions of our theory are robust as we have also highlighted earlier. We have added the following line in the revised manuscript:

Line 97-99 of the revised manuscript: By construction, each magic angle must be accompanied by two critical angles θ_{cr} (above and below θ_M), and therefore its effects should be robust to lattice relaxation effects (as we demonstrate explicitly below).

Referee Comment 12: Page 6, line 389, the authors mention using the relaxation time approximation in the Boltzmann equation. How well is the RTA justified in this case? For example, moving away from the Dirac points to the van Hove singularity, the dispersion is anisotropic, leading to tensor-like relaxation time. Or, what about possible density dependence of the relaxation time?

Although for historical reasons it is called the relaxation time approximation (RTA), our Boltzmann formalism for the electron-phonon scattering includes the density dependence of the relaxation time. Close to the van Hove singularity, the dispersion becomes anisotropic, and we keep the full anisotropy of the band structure when determining the Fermi velocity. However, for the transport scattering time, we do assume that it is isotropic. We have checked numerically that keeping the full anisotropic scattering time to compute the resistivity (ρ_{xx} and ρ_{yy}) gives indistinguishable results, justifying this approach. Moreover, we can physically understand this as a consequence of the thermal averaging step in the resistivity calculation. Finally, we take the geometric mean of the resistivity $\rho = \sqrt{\rho_{xx}\rho_{yy}}$ as is standard in the literature (see e.g. T. Ando, A. B. Fowler, and F. Stern, Rev. Mod. Phys. **54**, 437 (1982)). Since the full anisotropic transport calculation is computationally very expensive with no additional benefits, we use isotropic scattering time assumption in our work. For the case of the Planckian model (e.g. line 398 rather than line 389 of the original manuscript), by construction, it is assumed that the Planckian time has the isotropic form of $\hbar\tau_{pl}^{-1} = k_B T$ that is density independent (see e.g. Ref. [43], Nature **430**, 512 (2004), Phys. Rev. Lett. **94**, 111601 (2015), and Nat. Phys. **15**, 142 (2019)). In our work, we take this model as the starting assumption and do the Boltzmann transport theory for this model to get the temperature and density dependence of the resistivity.

Referee Comment 13: Page 7 Figure 5: The authors try to demonstrate that the parameters obtained from a fit to pre-existing experimental results agree better with the electron-phonon theory than with the Planckian model. Not all the cases portrayed are convincing. In panel c), the experimental parameters seem to show the opposite dependence on the

twist angle (decreasing) that the theoretical curves (increasing). Panel b) is hard to read due to the split and the different scales in the upper and lower plot. Moreover, the lower figure's experimental results do not seem to disagree too much from the theoretical results. I would strongly recommend revising this figure. One suggestion would be to plot relative deviations rather than absolute values to demonstrate agreement/disagreement across different scales. Moreover, it does not become clear from the main text why there is one panel showing data for "rigid (no geometry enhancement)" and to which calculation this curve corresponds.

We thank the referee for this suggestion. Thinking along these lines we experimented with different ways to present the data and settled on the new Figure 5 that follows the referee suggestion to show relative deviations as opposed to absolute values, and removes the data for "rigid (no geometry enhancement)", while incorporating both the atomistic and continuum relaxation models. The revised Figure. 5 is shown below.

The parameters obtained from fit to Refs. 2, 13, and 45 (see Extended Data for details). We find that the data agree much better with the expectations from the electron-phonon theory than the Planckian model. (a) Relative deviations of Fermi velocity as a function of twist angle obtained by fitting the experimental resistivity to the electron-phonon theory (b) obtained by fitting to the Planckian theory, (c) Relative deviations of effective electron-

phonon coupling constant $\tilde{\beta}_A$ and (d) Planckian strength C . The error bars represent 95% confidence interval of the parameters. The Planckian bound ($C \leq 1$) is violated for small twist angles. This rules out the Planckian model as the dominant transport mechanism in tBG.

The square represents the electrons while the circle represent the holes. The data points are horizontally shifted by 0.018° (-0.018°) for the electrons (holes) for clarity. The red (blue) color is the relative deviations of the experimental Fermi velocity or electron-phonon coupling constant resulted from the fitting with the theoretical calculation using relaxation model 1 (2). The In panel (a), (b), and (c), relative deviations up to 100 % is considered as a good agreement, while in panel (d), $0 \leq C \leq 1$ is considered as a good agreement as required by the Planckian bound.

Referee Comment 14: As a general comment. At length, the paper discusses and even reuses the results of ref 13, of which they argue in favour. Conversely, the results of ref 40, which they claim to disprove, are mentioned, but not discussed in detail. The manuscript would greatly benefit from a more balanced discussion of both sides.

We approached the authors of Ref. [42] asking them for their raw data in order for us to do an equal comparison. Although they responded favourably about giving us their data, they have yet to do so (despite reminders). We therefore tried our best to extract the data from the color plots in their papers and have included two data sets in our Figure 5. However, one issue is that the MIT group is very focused on the narrow density regime close to the superconducting regime, while our work is focused on the metallic regime that spans from charge neutrality until the van Hove singularity. It is easier for us to compare with Ref. [13] since they also are focusing on the same metallic regime. To avoid confusion, we have clarified the corresponding paragraph that introduces the experimental finding, i.e.

Line 162-165 of the revised manuscript: In this work, we focus on the metallic regime that is far from the superconducting regime. A detailed analysis ultimately shows that the dominant scattering mechanism in this regime is not Planckian for several reasons...

Remarks of Referee 2:

In the present manuscript Sharma et. al. address the transport phenomena (resistivity) in twisted bilayer graphene (TBLG) near the magic angle. They mostly considered the phononic contributions and compared their findings with experiments, and also contrasted them with the Planckian theory (Ref. 40). By solving the Boltzman transport equation at finite temperatures, the authors arrive at the conclusion that phonon driven transport phenomena is more consistent with the existing experimental observations, while the existing Planckian theory shows some discrepancies. The subject of this article is interesting and definitely of current interest due to the ongoing experimental works on TBLG near the magic angle. As far as I understand, the main punchline of this work is the suitability of phonon driven transport in TBLG near the magic angle to match with existing experiments, which otherwise cannot be reconciled by the Planckian mechanism. Therefore, I will judge the suitability of this article for the publication in Nature Communication based on this point. On this ground I bear reservation against the publication of this work (at least in the current format) for the following reasons.

We thank the referee for the comment. We address all of the referee's concerns below.

Major concerns of Referee 2:

Referee Comment 1: It is obvious from Fig. 2(b) that the Planckian theory is more consistent with experiments than the phonon theory. Specifically at high temperatures Planckian theory provides saturation of resistivity, which a pure phonon theory fails to capture.

Author Response 1: This is incorrect. It is simply not true that the pure phonon theory fails to capture the saturation. Indeed the saturation of the resistivity in the phonon theory done correctly is one of the main points of our paper! Perhaps the referee was confused thinking that only the dashed line in Fig 2(b) left panel was the phonon theory? Actually, all curves in that panel refer to the phonon theory and they clearly show a saturation. The dashed line is the phonon theory without the van Hove singularity, and it is precisely our point that this vHs must be included in the phonon theory. In part, this confusion might be because in the original submission, the figure label "phonon theory" intersected only with the dashed curve. We have added bounding boxes around the word "phonon theory" in Fig 2, to make it clear that this refers to all the curves, and not just the dashed curve. To further avoid any future confusion, we now explicitly discuss this a few times in the main text, e.g.

(i) Caption of Fig. 2 of the manuscript: "The electron-phonon scattering theory (left panels) correctly captures the (a) carrier density and (b) temperature dependence of experimentally observed resistivity (middle panels), unlike the Planckian theory (right panel) that shows a stronger density dependence. ... Solid lines in the electron-phonon and Planckian theory are for a two-band effective model that includes the van Hove Singularity, while the dashed lines are for the linear Dirac model. For electron-phonon scattering, the linear-in- T resistivity at low temperature is captured by the Dirac model, while the saturation at higher temperature requires the van Hove singularity."

(ii) Line 148-154 of the manuscript: “... They argue that the linear-in-temperature behaviour persisting well below the Bloch-Gruneisen temperature and the saturation of resistivity at higher temperature are both inconsistent with the conventional theory of phonon transport. We show here that both of these features are actually essential features of phonon-limited transport in tBG (see Fig. 1).”

(iii) Line 182-199 of the manuscript: “In Fig. 2 we compare data from Ref. 13 (middle panels) with both a phonon-limited theory (left-panel) and a Planckian theory (right panel). ... We note that both the phonon-limited theory and the Planckian theory are linear-in- T at low temperature, and saturate at high temperature (qualitatively similar to what is seen experimentally). However, the origin of the saturation is very different. For phonon scattering, the saturation is set by the electronic bandwidth $2\varepsilon_{\text{VHS}}$, while for Planckian dissipation it is mostly independent of ε_{VHS} and set by Planckian strength C , which is expected to be somewhat universal and $C \leq 1$.”

Referee Comment 2: Clearly, experimental data from Fig. 8 is not symmetric about zero doping. But phonon theory as well as Planckian theory produce symmetric curves for the resistivity.

Author Response 2: This is trivial issue. Electron-hole asymmetry arises from the second-nearest hopping in the effective tight-binding model for graphene. The superlattice potential renormalizes the kinetic energy scales including the asymmetry to lower energies, see e. g. R. Bistritzer and A. H. MacDonald, *Proc. Natl. Acad. Sci. USA* **108**, 12233 (2011). However, this effect is at most 14 percent of the total (see e.g. Fig. 8(b) middle panel). Rather than to worry about this, we chose to include this effect in the theory by fitting separately for electron and hole side (as seen by the red and blue points in Fig. 5). We have added this explanation in the manuscript.

Line 202-209 of the manuscript: Near magic-angle TBG has electron-hole asymmetry that arises from the second-nearest hopping in the effective tight-binding model for graphene. The superlattice potential renormalizes the kinetic energy scales including the asymmetry to lower energies [14]. We include this effect in the theory by fitting separately for electron and hole side (as seen by the red and blue points in Fig. 5)

Referee Comment 3: Also the phonon driven Boltzman transport has already been addressed in great details in PRB 93, 195103 (2016). Authors should definitely cite this work.

Author Response 3: We cite this paper in our revised version along with the other papers that discuss phonons in monolayer graphene (see Ref. [57]). For both graphene monolayers and bilayers, charged impurities dominate the electronic carrier transport at low temperature, acoustic phonons become relevant at intermediate temperatures ($T \sim 100K$), and at still higher temperatures ($T \sim 250K$), optical phonons take over as the dominant scattering mechanisms [see e.g. Rev. Mod. Phys. **83**, 407 (2011) for a complete discussion]. It is in this context that PRB **93**, 195103 (2016) discusses the role of optical phonons in monolayer graphene. This is not relevant to our current work on twisted bilayer graphene

since optical phonons are not dominant in any temperature regime we are interested in. This is in contrast to the acoustic phonons we have discussed in our work that dominate in most of the experimental temperature range. Nonetheless, we cite this reference as requested by the Referee.

Referee Comment 3b: In that regard, I do not see any novelty in the calculation, besides extending the existing analysis beyond the Dirac picture. Also it is not obvious from any figure, how important are the higher order in momentum terms (beyond Dirac picture). In order words, authors need to explicitly show the discrepancy between the results obtained from pure Dirac theory and beyond Dirac theory.

Author Response 3b: We respectfully and completely disagree with the referee on this point. We already addressed this point in the original submission and think the referee got confused by our figure labels as discussed above. For example, in the left panel of Fig. 2, the results of the Dirac theory for phonons are represented by the straight dashed line, while the results of ‘beyond the Dirac theory’ has the saturation at higher temperatures. **They are clearly different.** And therefore, what the referee calls the “higher order in momentum” or what we call the “van Hove singularities” are clearly essential to the saturation of resistivity. To avoid any future confusion, we include the following sentences in the main text:

Line 191-202 of the manuscript: “We note that both the phonon-limited theory and the Planckian theory are linear-in- T at low temperature, and saturate at high temperature (qualitatively similar to what is seen experimentally). However, the origin of the saturation is very different. For phonon scattering, the saturation is set by the electronic bandwidth $2\varepsilon_{\text{VHS}}$, while for Planckian dissipation it is mostly independent of ε_{VHS} and set by Planckian strength C , which is expected to be somewhat universal and $C \leq 1$. This illustrates that both the phonon-limited theory and the Planckian theory provide robust predictions that can be tested against experiment.”

Caption of Figure 2: “... Solid lines in the electron-phonon and Planckian theory are for a two-band effective model that includes the van Hove Singularity, while the dashed lines are for the linear Dirac model. For electron-phonon scattering, the linear-in- T resistivity at low temperature is captured by the Dirac model, while the saturation at higher temperature requires the van Hove singularity. For the Planckian theory, the Dirac model and the two-band model are quantitatively similar and show much stronger density dependence compared to experiment. In this case, the saturation at high-temperature is set not by the van Hove singularity, but by a universal value $\rho(T \rightarrow \infty) = C/(8 \ln 2) h/e^2$ (the coefficient $C \leq 1$ for Planckian dissipation). For most experimental data, including those showed here, $C \geq 1$. Taken together with the weak density dependence seen experimentally, this suggests that phonon scattering rather than Planckian dissipation is the dominant scattering mechanism at play in twisted bilayer graphene.”

Referee Comment 4: Also from Fig. 9, it is obvious that Planckian theory yields better agreement with experiment at least at low temperatures. Therefore, the present theory

fails to answer a simple question: Why phonon mediated transport then the absolute theory?

Author Response 4: To the contrary. It is obvious from Fig. 9 that the phonon theory yields better agreement. For example: (i) There is strong carrier density dependence in ALL the figures of the rightmost column (Planckian theory), but this is absent in the experiment (middle column) and the phonon theory (leftmost column); (ii) the value of the “ C ” needed to agree with experiment ranges from 0.5 to 9, while the Planckian theory requires $C < 1$. The disagreement is even stronger in Fig. 8 where the Planckian theory sharply peaked at charge neutrality (not seen in either the experiment or the phonon theory). We explicitly discuss this point in the main text:

Caption of Figure 2: “The electron-phonon scattering theory (left panels) correctly captures the (a) carrier density and (b) temperature dependence of experimentally observed resistivity (middle panels), unlike the Planckian theory (right panel) that shows a stronger density dependence. Experimental data is taken from Ref. [13] for $\theta = 1.11^\circ$ (comparison for devices with other twist angles is shown in the Extended Data). Solid lines in the electron-phonon and Planckian theory are for a two-band effective model that includes the van Hove Singularity, while the dashed lines are for the linear Dirac model. For electron-phonon scattering, the linear-in- T resistivity at low temperature is captured by the Dirac model, while the saturation at higher temperature requires the van Hove singularity. For the Planckian theory, the Dirac model and the two-band model are quantitatively similar and show much stronger density dependence compared to experiment. In this case, the saturation at high-temperature is set not by the van Hove singularity, but by a universal value $\rho(T \rightarrow \infty) = C/(8\ln 2)h/e^2$ (the coefficient $C \leq 1$ for Planckian dissipation). For most experimental data, including those showed here, $C \geq 1$. Taken together with the weak density dependence seen experimentally, this suggests that phonon scattering rather than Planckian dissipation is the dominant scattering mechanism at play in twisted bilayer graphene.”

Line 182-202 of the manuscript: “In Fig. 2 we compare data from Ref. [13] (middle panels) with both a phonon-limited theory (left-panel) and a Planckian theory (right panel). Similar to the experimental data, the phonon-mediated theory has weak density dependence. By contrast, the resistivity of the Planckian theory has strong density dependence (not seen in the experiment) that results from the density of states dependence of the Drude weight, which unlike electron-phonon, remains uncompensated by the scattering time. We note that both the phonon-limited theory and the Planckian theory are linear-in- T at low temperature, and saturate at high temperature (qualitatively similar to what is seen experimentally). However, the origin of the saturation is very different. For phonon scattering, the saturation is set by the electronic bandwidth $2\varepsilon_{\text{VHS}}$, while for Planckian dissipation it is mostly independent of ε_{VHS} and set by Planckian strength C , which is expected to be somewhat universal and $C \leq 1$. This illustrates that both the phonon-limited theory and the Planckian theory provide robust predictions that can be tested against experiment.”

Minor concerns of Referee 2:

Referee Comment 1: Authors say that a drop of resistivity by 5 orders is considered as a sign of superconductivity. They need to provide references for this statement. Presently it appears somewhat ad hoc.

Author Response 1: To address this issue, we add the following reference (where this issue is discussed):

'Symmetry breaking in twisted double bilayer graphene', Minhao He, Yuhao Li, Jiaqi Cai, Yang Liu, K Watanabe, T Taniguchi, Xiaodong Xu, Matthew Yankowitz, Nature Physics 1-5, 2020.

Referee Comment 2: Authors say that presently it is not clear whether the insulator is Mott insulator or Wigner crystal. The insulator can be a Slater insulator. Authors should mention it clearly.

Author Response 2: We add the following reference to address this issue:

'Magnetic Effects and the Hartree-Fock Equation', J. C. Slater. Phys. Rev. **82**, 538 (1951).

Referee Comment 3: Authors say that the role of the long range Coulomb interaction is not clear in TBLG. Note that TBLG is well described by slow Dirac fermions. For Dirac fermions it has been clearly established that long range Coulomb interactions can only lead to logarithmic increase of the Fermi velocity. See PRL **118**, 026403 (2017).

Author response 3: The statement that TBLG is described by slow Dirac fermions is only valid for a limited regime near charge neutrality (see Figure below). For higher densities relevant to the experimental data, this statement is incorrect. Indeed, extending the phonon calculation beyond the linear Dirac regime is one of the important tasks carried out in our manuscript. As seen in the figure [taken from Phys. Rev. B **99**, 140302(R) (2019)], the TBLG bandstructure is only described by Dirac fermions for a range of parameters. Since in the present work, we explore temperatures and densities close to the VHS (red line in the figure below), the Dirac theory (blue dashed line) is no longer valid.

Figure: *Left panel:* Electronic structure of twisted bilayer graphene for $\theta = 1.3^\circ$ is ~ 25 percent of the van Hove singularity energy (marked VHS). *Middle panel:* Phonons dominate the transport at temperatures higher than T_{cross} (black solid line), while charged

impurities dominate at lower temperatures. *Right panel:* Similarly, phonons dominate at high density (higher than the black solid line), while charged impurities dominate at low carrier density. (See Phys. Rev. B **99**, 140302(R) (2019) for details.)

For energies beyond the continuum Dirac model, we know that the role of lattice-scale effects become important – we discussed this issue for the case of monolayer graphene in Science **361** 570 (2018). Once the lattice scale is included, the statement that long-range Coulomb interactions *can only* lead to a logarithmic increase in the Fermi velocity is no longer correct. The case of TBLG is further complicated by the fact that the interaction energy becomes comparable or larger than the bandwidth. We discuss this complication in our introduction in order to make the connection to recent works e.g. Phys. Rev. B **98**, 235158 (2018) and PNAS **115**, 13174 (2018) where these issues are highlighted in further detail.

Referee Comment 4: Authors say Planckian theory in explaining experimental data from Ref. [13] violates the assumption for the validity of the Planckian theory. How?

Author Response 4: The Planckian assumption requires $C \lesssim 1$. This is established in the literature, for example see: [1] Nature **430**, 512 (2004) pg. 513 where they write “the laws of quantum physics forbid the dissipation time to be any shorter at a given temperature than [this Planckian time]” or [2] Phys. Rev. Lett. **94**, 111601 (2015) where they write “According to the uncertainty principle, the product of the energy of a quasiparticle ... and its mean free time ... cannot be smaller than \hbar , otherwise the quasi-particle concept does not make sense.” At high temperature, the energy of a quasiparticle is just the thermal energy $k_B T$; This can also be seen experimentally, see: [3] Fig. 2 of Science **339**, 804 (2013) or [4] Table. 1 of Nat. Phys. **15**, 142 (2019) that show experimentally a near universality of $C \approx 1$ (within $\lesssim 20$ percent) for a wide range of materials with two orders of magnitude variation in Fermi velocity. However, as seen in our Fig. 5d, the best fits to the experimental data of Ref. [13], violates this assumption of the Planckian bound with C ranging from 0.5 to 9.5. We now explicitly discuss this point in the main text of the manuscript:

Line 138-148 of the manuscript: By now there have been two experimental transport studies focusing on the metallic regime. The first is from the MIT group [42] and the second is a UCSB-Columbia collaboration [13]. While the two experiments are largely consistent with each other, they arrive at very different conclusions on the dominant scattering mechanisms at play. Ref. [42] argues for a Planckian mechanism to explain their data, which implies a scattering rate $\hbar\tau^{-1} = Ck_B T$, where $C \lesssim 1$ [43]. Here $C = 1$ is the Planckian bound set by holography and believed to be relevant for strange metals [44].

Caption of Fig. 2: For most experimental data, including those showed here, $C \geq 1$. Taken together with the weak density dependence seen experimentally, this suggests that phonon scattering rather than Planckian dissipation is the dominant scattering mechanism at play in twisted bilayer graphene.

Referee Comment 5: Authors claim that phonons “appear” to dominant transport. But,

it is not at all obvious from any figure. One can easily conclude that Planckian theory is equally good or bad.

Author Response 5: No. From the Figure 2, one can immediately see that the electron-phonon theory correctly captures the experimental features such as variation of resistivity with temperature and carrier density. Both the experimental data and the phonon mechanism show a weak density dependence and point out that the resistivity saturation at high temperature is set by the VHS energy. On the contrary, the Planckian theory does not correctly capture the experimental features as it predicts a stronger density dependence of resistivity that is not seen in experiments. Further, for the Planckian theory the saturation of resistivity is independent of the bandstructure. We explicitly discuss this point in the main text, e.g.

Caption of Figure 2: “The electron-phonon scattering theory (left panels) correctly captures the (a) carrier density and (b) temperature dependence of experimentally observed resistivity (middle panels), unlike the Planckian theory (right panel) that shows a stronger density dependence. Experimental data is taken from Ref. [13] for $\theta = 1.11^\circ$ (comparison for devices with other twist angles is shown in the Extended Data). Solid lines in the electron-phonon and Planckian theory are for a two-band effective model that includes the van Hove Singularity, while the dashed lines are for the linear Dirac model. For electron-phonon scattering, the linear-in- T resistivity at low temperature is captured by the Dirac model, while the saturation at higher temperature requires the van Hove singularity. For the Planckian theory, the Dirac model and the two-band model are quantitatively similar and show much stronger density dependence compared to experiment. In this case, the saturation at high-temperature is set not by the van Hove singularity, but by a universal value $\rho(T \rightarrow \infty) = C/(8\ln 2)h/e^2$ (the coefficient $C \leq 1$ for Planckian dissipation). For most experimental data, including those showed here, $C \geq 1$. Taken together with the weak density dependence seen experimentally, this suggests that phonon scattering rather than Planckian dissipation is the dominant scattering mechanism at play in twisted bilayer graphene.”

Line 164-181 of the manuscript: A detailed analysis ultimately shows that the dominant scattering mechanism is not Planckian for several reasons including: (a) the Planckian theory also predicts a strong carrier density dependence (absent in the experiment); (b) the experiment and the phonon mechanism both show the resistivity saturation at high temperature is set by the VHS energy, while for the Planckian theory this saturation is intrinsic (i.e independent of bandstructure); (c) the twist angle dependence of Fermi velocity as extracted from experiment for phonon-limited scattering is consistent with the continuum theory [14–16], while it is orders-of-magnitude off for the Planckian theory; and most significantly, (d) the extracted value of the scattering time from the experiment using the Planckian theory contradicts the assumptions of the Planckian theory. Our work shows that the phonon interpretation of Ref. [13] is consistent with the theory we develop here.

Line 182-202 of the manuscript: In Fig. 2 we compare data from Ref. 13 (middle panels)

with both a phonon-limited theory (left-panel) and a Planckian theory (right panel). Similar to the experimental data, the phonon-mediated theory has weak density dependence. By contrast, the resistivity of the Planckian theory has strong density dependence (not seen in the experiment) that results from the density of states dependence of the Drude weight, which unlike electron-phonon, remains uncompensated by the scattering time. We note that both the phonon-limited theory and the Planckian theory are linear-in- T at low temperature, and saturate at high temperature (qualitatively similar to what is seen experimentally). However, the origin of the saturation is very different. For phonon scattering, the saturation is set by the electronic bandwidth $2\varepsilon_{\text{VHS}}$, while for Planckian dissipation it is mostly independent of ε_{VHS} and set by Planckian strength C , which is expected to be somewhat universal and $C \leq 1$. This illustrates that both the phonon-limited theory and the Planckian theory provide robust predictions that can be tested against experiment.

Referee comment 6: Authors also report multiple (2 or 3) dips in the resistivity. But, I could not see any dip in resistivity in any experimental data.

Author response 6: The theory shown in Fig. 4 shows the hypothetical case of experiments measuring conductivity while continuously changing twist angle. While the MIT and Columbia group potentially have the capacity do this sort of experiment, such data has not yet appeared in the literature. (We know from private communication that both MIT and Columbia are planning to do such experiments, but as far as we know, these have not yet been done). This is why the Referee has not seen the dips in any experimental data. However, our Fig. 4 represents concrete predictions for experiments that are likely to be done in the near future.

By contrast, the experiment of Ref. [13] and Ref. [42] make samples with fixed angle and tune temperature and carrier density, and as seen in our Figs. 2, 8, and 9, there are no dips in the resistivity for such data.

Reviewers' Comments:

Reviewer #1:

Remarks to the Author:

The authors have addressed my questions and concerns in detail and in a convincing manner. The manuscript has been improved by further explanatory text and additional calculations and data. I am now convinced that the work "Carrier transport theory for twisted bilayer graphene in the metallic regime" will advance the general understanding of transport in twisted bilayer graphene in the metallic regime. Given the current great interest of the community in twisted materials I believe it is justified and timely to recommend the work for publication in Nature Communications.

Reviewer #2:

Remarks to the Author:

I thank the authors for clarifying the questions and concerns I raised in the first round of review. The authors have modified the text of the manuscript adequately to answer these questions. As I said previously, I do not undermine the importance of the calculation based on phonon driven scattering mechanism. However, the paper is being reviewed for publication in Nature Communications, which is a high impact journal of impact factor 12, much higher than Physical Review Letters. Therefore, besides presenting a solid calculation the authors need to make a convincing case that their work is sufficiently novel and supersedes pre existing theoretical works or framework. In that regard, I find that this manuscript fails to meet the high standards of Nature Communications for the following reasons.

1. As the authors agreed, which I also pointed out in the previous report, that the present manuscript is an extension of the existing phonon driven transport calculations in graphene or generally Dirac system, where only linear band dispersion has been accounted for. Specifically, authors now include the role of higher-gradient terms or van Hove singularity in the transport calculation. This statement is supported by the authors themselves, saying

"Indeed, extending the phonon calculation beyond the linear Dirac regime is one of the important tasks carried out in our manuscript."

While this is certainly a commendable effort, however, does not bring any novelty to the table. Simply because the formalism is well established for the Dirac system with only linear dispersion. Extending the existing formalism by including higher-gradient terms simply does not qualify as novel contribution.

2. The agreements of the phonon based computation of resistivity with experiments and its contrast with Plankian theory have been overblown. Take for example, Fig. 2 and Fig. 9. Both phonon based analysis and Plankian theory show carrier dependence of resistivity. Admittedly, Plankian theory shows "slightly stronger" dependence on carrier density than phonon theory. But, the differences between these two theoretical approaches is not by orders of magnitude. Thus comparison with experiments does not select one theory over the other.

3. Even more importantly, the experimental data from Ref. 13 have not been presented in an appropriate way. For example in Fig. 2(b) and Fig. 9, the "poinsize" of the experimental data is too large, which clearly overwhelms or masks any carrier density dependence. For instance, the "Red curves" are so "thick" that they sit over the other curves shown in blue, yellow, cyan for different carrier densities. This is certainly not how experimental data should be presented to claim the suitability of a particular theoretical approach or calculation. If one carefully looks through these experimental curves, taken from Ref. 13, there is clear carrier density dependence, which is qualitatively captured by both phonon and Plankian theory. Finally, Fig. 9 simply cannot pick one theory over the other. Finally, a "slightly" stronger carrier density dependence in the Plankian theory can easily be smeared out by disorder, which is definitely present in all these samples.

Therefore, I stick to my original decision. This paper stands as a good solid theoretical work, but does not meet the high standard bars for publication in Nature Communications. If, on the other hand, the paper contained original experimental data then purely based on the merit of

experiments, it could have been considered for Nature Communications. This is certainly not the author's fault. But, on the basis of private communications with experimental groups, who have not performed the experiment yet, I cannot assign any extra merit to this paper.

Hence, this paper should be published in a more specialized journal on condensed matter physics.

Reply to the Referees

Referee A: The authors have addressed my questions and concerns in detail and in a convincing manner. The manuscript has been improved by further explanatory text and additional calculations and data. I am now convinced that the work "Carrier transport theory for twisted bilayer graphene in the metallic regime" will advance the general understanding of transport in twisted bilayer graphene in the metallic regime. Given the current great interest of the community in twisted materials I believe it is justified and timely to recommend the work for publication in Nature Communications.

We thank the referee for their positive assessment of our manuscript, and for recommending its publication in Nature Communications.

Referee B: I thank the authors for clarifying the questions and concerns I raised in the first round of review. The authors have modified the text of the manuscript adequately to answer these questions. As I said previously, I do not undermine the importance of the calculation based on phonon driven scattering mechanism.

We thank the referee for taking the time to read our manuscript and detailed reply. We are happy that the referee is satisfied with all our answers from the first round of review.

However, the referee brings up **new** technical concerns in the second round of review, and we are happy for this opportunity to address them fully. We thank the referee in advance for their careful consideration of our arguments presented here.

We group the referee concerns into the following 5 topics that we address sequentially. As a result of this grouping, the points made by the referee are rearranged from their original order.

Summary of Referee B Concerns

1. Do the Planckian and phonon limited resistivities show similar carrier density dependence, and can experiment select one theory over the other?
2. Have we shown the experimental data appropriately?
3. Do we provide solid quantitative evidence to favor the phonon mechanism over the Planckian theory?
4. Can disorder smearing result in better quantitative agreement between the Planckian theory and experiment?
5. The novelty of our results.

It is our sincere hope that we are able to persuade the referee to our point of view on all five points listed above.

1. Do the Planckian and phonon limited resistivities show similar carrier density dependence, and can experiment select one theory over the other?

Referee B: Admittedly, Planckian theory shows "slightly stronger" dependence on carrier density than phonon theory. But, the differences between these two theoretical approaches is not by orders of magnitude. Thus comparison with experiments does not select one theory over the other.

Our Reply: The referee comment got us thinking – is there an easy way for us to demonstrate: (i) the Planckian theory and phonon theory have significantly different carrier density dependence, and (ii) Experiment really does select one theory over the other? Actually, we believe we can demonstrate this very easily. The figure below shows the same experimental data as Fig. 2 (for $T = 50\text{K}$) (top left panel), as well as 3 additional representative examples. To test the referee claim, we compare experimental data with both the phonon theory and Planckian theory taken as deviations from the value at $n = 0$. (As we explain in more detail in Point 3 below, the theory curves shown here are the same as before, except we use the best theoretical estimates for β_A and C). These new figures (which we add as Fig. 8 to the revised manuscript) make clear that the Planckian theory shows much stronger carrier density dependence than either the phonon theory or experiment. We thank the Referee for helping us make this point more clearly.

2. Have we shown the experimental data appropriately?

Referee B: Even more importantly, the experimental data from Ref. 13 have not been presented in an appropriate way. For example in Fig. 2(b) and Fig. 9, the "pointsize" of the experimental data is too large, which clearly overwhelms or masks any carrier density dependence. For instance, the "Red curves" are so "thick" that they sit over the other curves shown in blue, yellow, cyan for different carrier densities. This is certainly not how experimental data should be presented to claim the suitability of a particular theoretical approach or calculation.

Our Reply: We thank the referee for pointing out that the symbol point size might mask the carrier-density dependence or lack thereof. This was completely unintentional. On the next page and in the revised manuscript we replot the figure with a smaller point size. We would like to make two additional important points for the referee's consideration:

(a) We draw the referee's attention that we actually have two figures: (i) $\rho(n)$ to show the density dependence of resistivity at fixed temperature, and (ii) $\rho(T)$ for the temperature dependence at fixed density. It is much easier to determine the density dependence directly from $\rho(n)$, rather than looking at $\rho(T)$. More importantly, by showing both $\rho(n)$ and $\rho(T)$, our intent is to present the experimental data more completely.

(b) It occurs to us that perhaps the referee is concerned about the density dependence in the data for $T < 30$ K? We draw the referee's attention that a full blow up of $\rho(T)$ in this regime is shown in Fig. 7a, and $\rho(n)$ for this regime is directly shown in Fig. 6a. However, as we explain in Figs. 1, 6, and 7, and the main text (e.g. in the paragraph quoted below), this regime for $T < 30$ K is governed by a different scattering mechanism [which we previously addressed in PRB B 99, 140302 (2019)]. To avoid any future confusion, in the revised figure for the manuscript, we highlight this low temperature region with a bounding box and arrow to indicate that a blow up of this window can be found in Fig. 7a. Moreover, in the manuscript we write:

We also provide experimental verification (see Fig. 6 and 7 in Extended Data) of our previous predictions that charged impurity scattering takes over as the dominant scattering mechanism at very low temperatures (below 20K) and low carrier densities (below $\sim 10^{11}$ cm⁻²). Taken together with the present work on phonon scattering (that applies at high temperature and high carrier density), this now presents a complete theory for the carrier transport for twisted bilayer graphene.

3. Do we provide solid quantitative evidence to favor the phonon mechanism over the Planckian theory?

Referee B: The agreements of the phonon based computation of resistivity with experiments and its contrast with Planckian theory have been overblown.

Referee B: Finally, Fig. 9 simply cannot pick one theory over the other.

Our Reply: Qualitative disagreements over the interpretation of results, particularly comparing theory to experiment, are commonplace in science, and should not alone be grounds for rejection of the work. We remain confident about our conclusions and outline here in some detail our methodology. So how should one decide the agreement between theory and experiment? One approach is to use the best theoretical estimates and compare with experiment. This approach is good when looking at a single data set with a single variable. We showed this approach above in Point 1 (and in the new Fig. 8). Comparing theory with two representative data sets each from MIT and UCSB-Columbia for $\Delta\rho(n)$ in our view clearly shows that the phonon theory works better.

Another standard approach is to use fitting parameters and then judge the reasonableness of the obtained best-fit parameters with theoretical estimates. We used the latter approach in our manuscript because we look at 14 data sets, each with two variables (density and temperature) and compared to two different theoretical models for lattice relaxation.

Given the complexity of our analysis, the only way to make any judgement on agreement between theory and experiment (and in particular if our claims are overblown) is to look at Fig. 5 where we compare “The parameters obtained from fit to Refs. 2, 13, and 60” and find “that the data agree much better with the expectations from the electron-phonon theory than the Planckian model”.

Let us try to make this point even clearer: what if fitting experiment to one theory required the electron velocity to be larger than the speed of light? This is clearly unphysical and would invalidate the theory. However, this invalidation of the theory would show up in Fig. 5, not in Fig. 2. In fact, as we already discussed in our first reply, we have exactly such a situation here. The Planckian theory requires $C \leq 1$. The Planckian theory fit in Fig. 2 gives $C = 2.0 \pm 0.1$ (and for some data sets, the value of the “C” needed to agree with experiment is as large as 9, well beyond the $C \leq 1$ criteria for the Planckian theory to be valid). This can be seen clearly in Fig. 5:

Any discussion about agreement between theory and experiment is incomplete without considering Fig. 5 which we developed following the recommendations of Referee A. It represents a comparison of 14 samples from two different experimental groups, with each sample consisting of a 2D grid of data in the density-temperature plane. Notice:

(i) **The top two panels** compare on equal footing the difference between the best fit Fermi velocity for the phonon theory and the Planckian theory and theoretical expectations. For the phonon theory, ALL data points lie inside the “good agreement” window, while for the “Planckian theory”, the data points with $\theta > 1.59^\circ$ (representing 30 percent of the data) all lie outside this window.

(ii) **The bottom two panels** compare the parameter for the phonon theory β_A and the parameter for the Planckian theory C with theoretical expectations. For the phonon theory 85 percent lie in the “good agreement” window, while the remaining cases are borderline. In contrast, for the Planckian theory only 15 percent lie clearly in the “good agreement” window.

(iii) **Taken together**, for the “Planckian theory”, there is not a single data set for which both parameters lie within “good agreement” window.

Having outlined our methodology to quantitatively compare the two theories, we convincingly conclude the phonon theory is better. This is in addition to any “goodness of fit” considerations discussed in Point 1 above.

4. Can disorder smearing result in better quantitative agreement between the Planckian theory and experiment?

Referee B: Take for example, Fig. 2 and Fig. 9. Both phonon based analysis and Planckian theory show carrier dependence of resistivity

Referee B: If one carefully looks through these experimental curves, taken from Ref. 13, there is clear carrier density dependence, which is qualitatively captured by both phonon and Planckian theory.

We do not dispute the fact that the experimental data does show some density dependence (which is why in Fig. 8 – now Fig. 9 in the revised manuscript – we show 24 different experimental data traces for $\rho(n)$) or that both the phonon and Planckian theory show “qualitative” agreement with experiment. We specifically discuss this in the manuscript:

We note that both the phonon-limited theory and the Planckian theory are linear-in- T at low temperature, and saturate at high temperature (*qualitatively* similar to what is seen experimentally). ... We show here that a Planckian mechanism also gives a saturation in resistivity at high temperature and linear-in-temperature behaviour at low temperature *consistent with experimental observations*.

However, we stand by the claim that our quantitative analysis picks the phonon theory over the Planckian theory.

A detailed analysis ultimately shows that the dominant scattering mechanism in this regime is not Planckian for several reasons including: (a) the Planckian theory also predicts a strong carrier density dependence (absent in the experiment); (b) the experiment and the phonon mechanism both show the resistivity saturation at high temperature is set by the VHS energy, while for the Planckian theory this saturation is intrinsic (i.e independent of bandstructure); (c) the twist angle dependence of Fermi velocity as extracted from experiment for phonon-limited scattering is consistent with the continuum theory, while it is orders-of-magnitude off for the Planckian theory; and most significantly, (d) the extracted value of the scattering time from the experiment using the Planckian theory contradicts the assumptions of the Planckian theory.

Referee B: Finally, a “slightly” stronger carrier density dependence in the Planckian theory can easily be smeared out by disorder, which is definitely present in all these samples.

We can easily test the effect of disorder on the Planckian theory and thank the referee for this suggestion – since in principle it is possible that the stronger carrier density dependence in the Planckian theory could be smeared out by disorder. In the figure below (included in the revised manuscript), following the methods discussed in our review article for tackling smearing by disorder in graphene-like systems [Reviews of Modern Physics 83, 407 (2011)], we show the effect of smearing by disorder using the effective medium theory (EMT) with density fluctuations n_{rms} consistent with values from Fig. 6. As can be seen, in practice, for realistic values of disorder, the smearing does not make the Planckian theory agree any better with experiment.

Figure. 3 Same plot as in Fig. 1, but now with disorder smearing (EMT) shown for the Planckian theory. Disorder smearing does not significantly change the Planckian curve.

5. The novelty of our results.

Referee B: However, the paper is being reviewed for publication in Nature Communications, which is a high impact journal of impact factor 12, much higher than Physical Review Letters. Therefore, besides presenting a solid calculation the authors need to make a convincing case that their work is sufficiently novel and supersedes pre existing theoretical works or framework. In that regard, I find that this manuscript fails to meet the high standards of Nature Communications for the following reasons. 1. As the authors agreed, which I also pointed out in the previous report, that the present manuscript is an extension of the existing phonon driven transport calculations in graphene or generally Dirac system, where only linear band dispersion has been accounted for. Specifically, authors now include the role of higher-gradient terms or van Hove singularity in the transport calculation. This statement is supported by the authors themselves, saying "Indeed, extending the phonon calculation beyond the linear Dirac regime is one of the important tasks carried out in our manuscript." While this is certainly a commendable effort, however, does not bring any novelty to the table. Simply because the formalism is well established for the Dirac system with only linear dispersion. Extending the existing formalism by including higher-gradient terms simply does not qualify as novel contribution.

Referee B: Therefore, I stick to my original decision. This paper stands as a good solid theoretical work, but does not meet the high standard bars for publication in Nature Communications. If, on the other hand, the paper contained original experimental data then purely based on the merit of experiments, it could have been considered for Nature Communications. This is certainly not the author's fault. But, on the basis of private communications with experimental groups, who have not performed the experiment yet, I cannot assign any extra merit to this paper. Hence, this paper should be published in a more specialized journal on condensed matter physics.

Our Reply: The referee focuses on one aspect of our work: extending the acoustic phonon scattering theory to beyond the Dirac regime to include the effects of the van Hove singularity. Even just in this respect, our view is that this represents the first major development in the theory for acoustic phonon scattering in graphene-like systems in more than a decade. More importantly, we would like to draw the referee's attention to some of the other major accomplishments in our manuscript:

(i) As we explain starting on line 268, by doing the full finite-temperature dynamic screening theory for the first time, and backed up with Figures S3, S4, and S5 of the supplemental material, we show conclusively that the deformation potential is screened. This resolves 30-year old debate on whether or not the deformation potential is screened.

(ii) As we explain starting on line 295, we establish yet another special angle in twisted moire systems: θ_{CR} , the angle at which the Fermi velocity equals the phonon velocity. For twisted bilayer graphene $\theta_{CR} \approx 1.15$ degrees. We show that at this angle, the phonon contribution to the resistivity strictly vanishes, and the experimentally measured resistivity would increase by several orders of magnitude for small deviations in angle on either side

of θ_{CR} . This is a concrete prediction that can be tested in experiments.

(iii) As we explain starting on line 325, we go beyond just intra-band kinematics and include inter-band scattering that becomes relevant for twisted bilayer graphene. We show for the first time that due to inter-band scattering, the linear-in-T resistivity can persist to temperatures much lower than the Bloch-Gruneisen temperature (since it is set instead by the Fermi temperature). This is explained in Fig. 3.

(iv) Starting on line 477, we calculate for the first time the density and temperature dependence of the resistivity for Planckian model showing that the slope of resistivity with temperature in the low temperature regime is set by both v_{F} and C , while the saturation of resistivity with temperature is set only by C . Equations 8, 9 and 10 for the Planckian theory are original to our work.

We hope the referee is persuaded that these are significant theoretical developments, and the manuscript is suitable for publication in Nature Communications.

Reviewers' Comments:

Reviewer #2:

Remarks to the Author:

In the rebuttal and revised manuscript the authors made sincere attempts to address my criticisms once again. First of all, I want to assure the authors that in the second round of review I did not raise any fundamentally new objections or criticisms. Most of them emphasized my previous concerns. While I appreciate the effort invested by the authors and the importance of the theoretical analysis, I stand by my previous judgement that this manuscript does not meet the high standards of Nature Communications.

One of the main objections I had concerns the novelty of this work. Over the review process it became more and more evident that the theoretical analysis of this manuscript extends the existing ones in the literature by incorporating higher-gradient terms, which in turn incorporate the effects of van Hove singularity in twisted bilayer graphene. As I said previously that these types of calculations are important and enrich our understanding of transport phenomena in interacting (via e-e or e-ph) systems, I cannot see the appearance of any new concept or methodological advancement. All the necessary tools are readily available and the majority of the calculations are already present in the literature. We may find in the future a new type of Dirac system where the higher gradient terms are different (such as the ones in higher-order van Hove singularity, see Phys. Rev. Research 1, 033206 (2019), which can be relevant in twisted bilayer graphene). A repetition of this analysis for higher-order van Hove singularity should not lead to a publication in Nature Communications. Authors may find my standards for novelty a bit harsh, but this is always a subjective matter and varies from person to person.

As far as the agreement with experiments is concerned, as shown in Fig. 8, that depends on the experimental data set. For example, the phonon theory matches reasonably well with the data from Refs. 2 and 13, but the agreement with the data from Ref. 13 and 60 is not at all good. And effective medium theory is not the only approach to incorporate disorder effects in Plancian theory. The fitting parameters β_A and C do not show any relation with the twist angle, i.e., they are neither monotonically increasing or decreasing with the twist angle away from the magic angle.

Fig. 5 also presents a comparative study of the phonon theory and Plancian theory. I agree that phonon theory shows better agreement with experimental data, falling mostly within the gray "Good agreement" region. However, the Plancian theory is not far apart. For example, the Fermi velocity plot (top panel, right) shows "Good agreement" with experiment with only one datum falling outside this regime. On the other hand, Plancian theory shows good agreement with resistivity data in Fig. 2(b).

One can go into such minuscule details to compare two theories. However, if one mechanism clearly wins over the other, such detailed comparison will not be needed. While I do not bear doubt that phonon theory has some regime of validity in twisted bilayer graphene, Plancian theory is not completely inoperative. Therefore, I object to the use of the phrase "... this now presents a complete theory for the carrier transport for twisted bilayer graphene". I personally think that this issue is far from being settled. The present manuscript definitely makes an important advancement in this direction, but it still lacks the level of novelty and impact to guarantee publication in Nature Communication.

Changes to the Manuscript

1. Following the editorial guidelines, we have now divided the manuscript into the following sections: Introduction, Results and Discussion (with subsections) and Methods.
2. Nature Communications does not have an Extended Data section. We separated our Extended Data section, putting most of it in the Methods section, but some in the Supplementary Material. The parts of the main text that referred to the Extended Data was modified appropriately.
3. As per the suggestion of the Referee, we have changed the last sentence of the Discussion section to "...the present work on phonon scattering (that applies at high temperature and high carrier density), this now presents a complete theory for the carrier transport for twisted bilayer graphene *in the metallic regime*."

Reply to third report of Referee 2

Referee Comment 1: In the rebuttal and revised manuscript the authors made sincere attempts to address my criticisms once again. First of all, I want to assure the authors that in the second round of review I did not raise any fundamentally new objections or criticisms. Most of them emphasized my previous concerns.

Author Reply 1: We thank the Referee for taking the time to look at our manuscript a third time and to provide a third referee report. We note that none of the objections or criticisms of the Referee in this report or any previous report are about the technical correctness of the work, which are not in question.

Referee Comment 2: While I appreciate the effort invested by the authors and the importance of the theoretical analysis, I stand by my previous judgement that this manuscript does not meet the high standards of Nature Communications.

One of the main objections I had concerns the novelty of this work. Over the review process it became more and more evident that the theoretical analysis of this manuscript extends the existing ones in the literature by incorporating higher-gradient terms, which in turn incorporate the effects of van Hove singularity in twisted bilayer graphene. As I said previously that these types of calculations are important and enrich our understanding of transport phenomena in interacting (via e-e or e-ph) systems, I cannot see the appearance of any new concept or methodological advancement. All the necessary tools are readily available and the majority of the calculations are already present in the literature. We may find in the future a new type of Dirac system where the higher gradient terms are different (such as the ones in higher-order van Hove singularity, see Phys. Rev. Research 1, 033206 (2019), which can be relevant in twisted bilayer graphene). A repetition of this analysis for higher-order van Hove singularity should not lead to a publication in Nature Communications. Authors may find my standards for novelty a bit harsh, but this is always a subjective matter and varies from person to person.

Author Reply 2: As the Referee points out, issues of novelty are somewhat subjective. On this issue, our view is closer to Referee 1 who wrote: “I am now convinced that the work ‘Carrier transport theory for twisted bilayer graphene in the metallic regime’ will advance the general understanding of transport in twisted bilayer graphene in the metallic regime. Given the current great interest of the community in twisted materials I believe it is justified and timely to recommend the work for publication in Nature Communications”.

Referee Comment 3: As far as the agreement with experiments is concerned, as shown in Fig. 8, that depends on the experimental data set. For example, the phonon theory matches reasonably well with the data from Refs. 2 and 13, but the agreement with the data from Ref. 13 and 60 is not at all good.

Author Reply 3: We agree with the Referee that the agreement between theory and experiment is better for some samples than for others. This is inevitable. However, the point of Fig. 8 was to see which theory, phonon or Planckian agrees better with experiment. For each of the four representative samples shown, the phonon theory is significantly closer to the experimental data than the Planckian theory.

Referee Comment 4: And effective medium theory is not the only approach to incorporate disorder effects in Plancian theory.

Author Reply 4: There are two broad classes of approaches to incorporate disorder into any transport theory. The first class is to assume carrier density inhomogeneity in real space, and the second class is to assume disorder broadening of the quasiparticles. While the two classes give significant qualitative differences, within each class the various methods give results that are qualitatively similar with only small quantitative differences. We know that for two-dimensional materials, the density inhomogeneity effects strongly dominate over quasiparticle broadening. And within the class of approaches that incorporate carrier density inhomogeneity, they all will give qualitatively similar results with small quantitative differences. While we believe that the effective medium theory gives the most accurate results, the figure would not change much if we used a two-carrier model, for example. All of this is discussed in some detail in Rev. Mod. Phys. 83 407 (2011) where we show successful the EMT approach has been in understanding experiments in disordered 2D materials.

Referee Comment 5: The fitting parameters β_A and C do not show any relation with the twist angle, i.e., they are neither monotonically increasing or decreasing with the twist angle away from the magic angle.

Author Reply 5: We do not understand what the Referee means here. $C = 1$ is a constant and therefore should not depend on twist angle. While β_A does depend on twist angle as we show in Figure S7 of the (original) supplementary material. Perhaps the Referee missed the fact that in Fig. 5c we plot the difference between β_{Afit} and $\beta_{Atheory}$? If $\beta_{Afit} - \beta_{Atheory}$ shows no dependence on twist angle, and $\beta_{Atheory}$ depends on twist angle, then it logically follows that β_{Afit} depends on twist angle.

Referee Comment 6: Fig. 5 also presents a comparative study of the phonon theory and Planckian theory. I agree that phonon theory shows better agreement with experimental data, falling mostly within the gray "Good agreement" region. However, the Planckian theory is not far apart. For example, the Fermi velocity plot (top panel, right) shows "Good agreement" with experiment with only one datum falling outside this regime. On the other hand, Planckian theory shows good agreement with resistivity data in Fig. 2(b).

Author Response 6: This statement by the Referee contains some inaccuracies that we clarify below. Moreover, we have addressed this point in our previous reply. Nevertheless, we repeat our argument here for completeness. Figure 5 represents a comparison of 14 samples from two different experimental groups, with each sample consisting of a 2D grid of data in the density-temperature plane. Please notice that:

(i) The top two panels compare on equal footing the difference between the best fit Fermi velocity for the phonon theory and the Planckian theory and theoretical expectations. For the phonon theory, ALL data points lie inside the "good agreement" window, while for the "Planckian theory", the data points with $\theta > 1.59^\circ$ lie outside this window. To be clear: 4 samples out of 14 are close to 30 percent of the full data set, and not just "one datum".

(ii) The bottom two panels compare the parameter for the phonon theory β_A and the parameter for the Planckian theory C with theoretical expectations. For the phonon theory, 85 percent lie in the "good agreement" window, while the remaining cases are borderline. In contrast, for the Planckian theory only 15 percent lie clearly in the "good agreement" window.

(iii) Taken together, for the "Planckian theory", there is not a single data set for which both parameters lie within "good agreement" window.

Having outlined our methodology to quantitatively compare the two theories, we convincingly conclude the phonon theory is better. This is in addition to any "goodness of fit" considerations seen in Figure 2b. It is simply not true that the Planckian theory (third panel in Fig. 2b) has better agreement than the phonon theory (first panel in Fig. 2b). It can clearly be seen that the experiment (middle panel in Fig. 2b) in the temperature regime of interest ($50\text{K} < T < 150\text{K}$) shows almost no temperature dependence as does the phonon theory, while the Planckian theory shows strong density dependence. This fact can be seen even more clearly in Fig. 2a e.g. the blue curve for $T = 50\text{K}$ is a flattish line in the left and middle panel, but shows strong density dependence in the right panel.

Referee Comment 7: One can go into such minuscule details to compare two theories. However, if one mechanism clearly wins over the other, such detailed comparison will not be needed.

Author Reply 7: Actually, it is pretty evident e.g. Fig. 2a, Fig. 8, Fig. 5, that one mechanism clearly wins over the other. We strongly disagree with the characterization that we are comparing minuscule details.

Referee Comment 8: While I do not bear doubt that phonon theory has some regime

of validity in twisted bilayer graphene, Plancian theory is not completely inoperative. Therefore, I object to the use of the phrase "... this now presents a complete theory for the carrier transport for twisted bilayer graphene". I personally think that this issue is far from being settled. The present manuscript definitely makes an important advancement in this direction, but it still lacks the level of novelty and impact to guarantee publication in Nature Communication.

Author Comment 8: We thank the Referee for catching the typo in the our last sentence. We did not intend to suggest that we have a complete theory for carrier transport for twisted bilayer graphene in all regimes. As indicated by our title, our focus is narrowly on the metallic regime. We have changed this sentence to read "... this now presents a complete theory for the carrier transport for twisted bilayer graphene in the metallic regime".